# Osteocytes directly regulate osteolysis via MYD88 signaling in bacterial bone infection

Tetsuya Yoshimoto[1,2], Mizuho Kittaka[1,2], Andrew Anh Phuong Doan[1,2], Rina Urata [1,2], Matthew Prideaux[2,3], Roxana E. Rojas[4], Clifford V. Harding [5], W. Henry Boom[5,6,7], Lynda F. Bonewald [2,3], Edward M. Greenfield[2,3,8] & Yasuyoshi Ueki [1,2] ✉

The impact of bone cell activation on bacterially-induced osteolysis remains elusive. Here, we show that matrix-embedded osteocytes stimulated with bacterial pathogen-associated molecular patterns (PAMPs) directly drive bone resorption through an MYD88-regulated signaling pathway. Mice lacking MYD88, primarily in osteocytes, protect against osteolysis caused by calvarial injections of bacterial PAMPs and resist alveolar bone resorption induced by oral *Porphyromonas gingivalis (Pg)* infection. In contrast, mice with targeted MYD88 restoration in osteocytes exhibit osteolysis with inflammatory cell infiltration. In vitro, bacterial PAMPs induce significantly higher expression of the cytokine RANKL in osteocytes than osteoblasts. Mechanistically, activation of the osteocyte MYD88 pathway up-regulates RANKL by increasing binding of the transcription factors CREB and STAT3 to *Rankl* enhancers and by suppressing K48-ubiquitination of CREB/CREB binding protein and STAT3. Systemic administration of an MYD88 inhibitor prevents jawbone loss in *Pg*-driven periodontitis. These findings reveal that osteocytes directly regulate inflammatory osteolysis in bone infection, suggesting that MYD88 and downstream RANKL regulators in osteocytes are therapeutic targets for osteolysis in periodontitis and osteomyelitis.

Host-microbiota interactions play fundamental roles in the activation of the innate immune systems. However, while dysregulation of the interaction is responsible for a wide range of immune-mediated diseases[1], the impact of bone cell-microbiota interaction on the skeleton and immune cell activation is poorly understood. Osteocytes are long-lived, multifunctional, and the most abundant bone cells embedded in the bone matrix. They are terminally differentiated cells derived from bone-forming osteoblasts on the bone surface[2]. In osteomyelitis, it is reported that *Staphylococcus aureus (S. aureus)* colonizes in the osteocyte lacunar-canalicular system of murine and human bones[3–5], implying that bacteria could directly interact with osteocytes within the bone infected by bacteria. It remains unknown, however, whether bacterial pathogens directly stimulate osteocytes in vivo and, if so, what type of innate immune receptors on osteocytes are used to respond to the specific bacterial pathogen-associated molecular patterns (PAMPs). Furthermore, while osteocytes are known

[1]Department of Biomedical Sciences and Comprehensive Care, Indiana University School of Dentistry, Indianapolis, IN 46202-5126, USA. [2]Indiana Center for Musculoskeletal Health, Indiana University School of Medicine, Indianapolis, IN 46202-5126, USA. [3]Department of Anatomy, Cell Biology, and Physiology, Indiana University School of Medicine, Indianapolis, IN 46202-5126, USA. [4]Janssen Biopharma Inc, Brisbane, CA 94005-1809, USA. [5]Department of Pathology, Case Western Reserve University & University Hospitals Cleveland Medical Center, Cleveland, OH 44106-4960, USA. [6]Department of Medicine, Case Western Reserve University & University Hospitals Cleveland Medical Center, Cleveland, OH 44106-4960, USA. [7]Department of Molecular Biology and Microbiology, Case Western Reserve University & University Hospitals Cleveland Medical Center, Cleveland, OH 44106-4960, USA. [8]Department of Orthopaedic Surgery, Indiana University School of Medicine, Indianapolis, IN 46202-5126, USA. ✉e-mail: uekiy@iu.edu

to produce a variety of cytokines and signaling molecules in response to outside stimuli[6,7], the in vivo impact of osteocyte-derived inflammatory mediators on bone homeostasis and immune cell regulation remains largely unidentified. On the other hand, previous studies of osteocyte-selective receptor-activator of nuclear factor-κB ligand (RANKL, encoded by the *Tnfsf11* gene)-deficient mice demonstrated that one of the essential features of osteocytes is the capacity to control osteoclastogenesis by directly providing RANKL to osteoclast precursors on the bone surface during the process of bone remodeling[8–10]. The importance of osteocyte-derived RANKL for bone destruction has also been suggested in a murine model of periodontitis[11], but the receptors on osteocytes and downstream signaling pathways through which oral bacteria induce RANKL expression to cause alveolar bone osteolysis have never been identified.

Toll-like receptors (TLRs) are a family of pattern recognition receptors involved in the initial phase of host defense against invading bacterial pathogens[12]. Myeloid differentiation primary response 88 (MYD88) is an essential adaptor protein for the downstream signaling of all TLRs except for TLR3. Activation of the TLR-MYD88 pathway signals the downstream NF-kB and MAPK pathways, resulting in the production of pro-inflammatory cytokines[13]. Previous cell culture studies showed that stimulation of TLR2 and TLR4 by bacterial PAMPs induces RANKL expression in osteoblastic cells[14–17]. However, the in vivo role of the TLR-MYD88 signaling pathway in osteoblast lineage cells for the regulation of osteoclast formation and bone resorption remains undetermined. As a result, pathological consequences of osteocyte activation in bone infections have never been studied in vivo. In addition, underlying molecular mechanisms by which the TLR-MYD88 pathway regulates RANKL expression in osteocytes are poorly understood. Therefore, revealing the impact of bacterially-activated osteocytes on the skeleton and identifying the molecular pathway of bacterially-activated osteocytes leading to RANKL expression could provide significant implications for treating osteolysis associated with bone infections such as periodontitis and osteomyelitis because osteocytes are the most abundant bone cells predominantly producing RANKL[8–10]. Furthermore, discovering the novel molecules regulating RANKL expression in osteocytes might lead to new treatment strategies for excessive bone resorption resulting from increased osteoclastogenesis, as occurs in osteoporosis and rheumatoid arthritis.

Here, we show that targeted deletion of MYD88, predominantly in osteocytes, fully protects against bone destruction in PAMPs-driven osteolysis models in mice. Remarkably, selective restoration of MYD88 in osteocytes is sufficient to cause osteolysis and inflammation in the models. In vitro, osteocytes have a significantly greater capacity to express RANKL than their precursor cells, osteoblasts, when stimulated with TLR2 and TLR4 agonists. Activation of the MYD88 pathway induces RANKL expression in osteocytes by activating cAMP responsive element binding protein (CREB) and signal transducer and activator of transcription 3 (STAT3) and enhancing the protein stability of these transcription factors. The E3 ubiquitin ligases F-box and leucine-rich repeat protein 19 (FBXL19) and PDZ and LIM domain 2 (PDLIM2) are involved in the mechanism of CREB/CREB binding protein (CBP) and STAT3 protein degradation, respectively. Translationally, administration of an MYD88 inhibitor protects against alveolar bone loss in mice orally infected with *Porphyromonas gingivalis* (*Pg*). These findings indicate that osteocytes are critical bacterial sensors in the bone and directly regulate osteolysis by integrating the MYD88 signaling into the RANKL regulatory mechanism in bone infection. Thus, the current study reveals a function of the MYD88 pathway in the skeletal system and provides a genetic basis for developing new approaches for treating infectious osteolysis, which target the MYD88 and downstream RANKL-regulating molecules in osteocytes.

## Results

### Targeted deletion of MYD88 in osteocytes and mature osteoblasts protects against calvarial osteolysis induced by PAMPs

To determine if activation of the TLR2/4-MYD88 signaling in osteocytes and mature osteoblasts impacts bone resorption caused by PAMPs, we generated osteocyte/mature osteoblast-selective MYD88 knockout mice using the dentin matrix protein 1 (*Dmp1*)-Cre promoter following the confirmation of TLR2, TLR4, and MYD88 expression in these cells (Supplementary Fig. 1). We also confirmed that *Dmp1-Cre* causes no obvious off-target *Myd88* gene deletion in immune and hematopoietic tissues that can be the source of RANKL and inflammatory mediators (Supplementary Fig. 2).

To induce osteolysis, Pam3CSK4 (hereafter referred as Pam3) or *Escherichia coli* (*E. coli*)-derived lipopolysaccharides (LPS) were injected onto the calvaria of *Dmp1-Cre;Myd88[fl/fl]* mice. MicroCT analysis revealed that the *Dmp1-Cre;Myd88[fl/fl]* mice exhibited a significant decrease in osteolysis compared to *Myd88[fl/fl]* mice in both sexes (Fig. 1a, b and Supplementary Fig. 3a, b). TRAP staining showed a decrease in osteoclast numbers in *Dmp1-Cre;Myd88[fl/fl]* mice (Fig. 1c, d and Supplementary Fig. 3c, d). Expression levels of *Rankl* were downregulated in the bone tissue from *Dmp1-Cre;Myd88[fl/fl]* mice, while osteoprotegerin (OPG, encoded by the *Tnfrsf11b* gene) levels were comparable, resulting in the decrease of *Rankl/Opg* ratio, a parameter for assessing osteoclastogenesis in the bone (Fig. 1e and Supplementary Fig. 3e). Pam3 decreased *Opg* levels in the bone regardless of the presence or absence of MYD88 deletion (Fig. 1e). Moreover, *Dmp1-Cre;Myd88[fl/fl]* mice were protected against osteolysis with reduced osteoclastogenesis when they were challenged with *S. aureus*-derived lipoteichoic acid (LTA), *Pg*-derived LPS, or live *Pg* (Supplementary Fig. 4a–h, Supplementary Fig. 5a–d). RANKL deletion in osteocytes and mature osteoblasts of *Dmp1-Cre;Rankl[fl/fl]* mice blocked bone erosion and osteoclast formation in calvarial injection models (Supplementary Fig. 6a–h). In contrast, deletion of interleukin-1 receptor (IL-1R) in *Dmp1-Cre;Il1r1[fl/fl]* mice did not rescue bone erosion or increased osteoclastogenesis and *Rankl* expression (Supplementary Fig. 7a–j). Collectively, these results show that activation of the TLR2/4-MYD88-RANKL axis in osteocytes and mature osteoblasts is responsible for osteolysis induced by PAMPs. No significant changes in bone properties of the femur of *Dmp1-Cre;Myd88[fl/fl]* mice under physiological conditions (Supplementary Fig. 8) indicated that the osteocyte/mature osteoblast MYD88 pathway is important for bone resorption when the pathway is bacterially activated.

### Targeted deletion of MYD88 in osteocytes and mature osteoblasts protects against alveolar bone resorption in periodontitis

T and B lymphocytes are well-known cellular sources of RANKL in periodontitis[18,19]. Therefore, the role of these cells in alveolar bone loss induced by oral *Pg* infection was investigated. Remarkably, RAG1-deficient mice (*Rag1[−/−]*) that lack T/B lymphocytes showed no significant protection against alveolar bone loss (Fig. 2a). We next examined if the TLR2-MYD88 axis in osteocytes and mature osteoblasts directly controls *Pg*-induced alveolar bone loss because TLR2 signaling is known to be activated in the *Pg* periodontitis model[20,21] and alveolar bone osteocytes/mature osteoblasts express TLR2 and MYD88 (Supplementary Fig. 9a). We found that both *Dmp1-Cre;Tlr2[fl/fl]* and *Dmp1-Cre;Myd88[fl/fl]* mice infected with *Pg* exhibited decreases in the distance between cementoenamel junction (CEJ) and alveolar bone crest (ABC) and alveolar loss compared to *Pg*-infected *Tlr2[fl/fl]* and *Myd88[fl/fl]* mice, respectively (Fig. 2b, c). The decreases were accompanied by suppressed osteoclast formation on the alveolar bone surface (Fig. 2d, e). Also, *Rankl* levels in alveolar bone tissue were decreased in *Pg*-infected *Dmp1-Cre;Tlr2[fl/fl]* and *Dmp1-Cre;Myd88[fl/fl]* mice (Supplementary Fig. 9b, c). Supporting a previous report[11], alveolar bone loss and increased osteoclast formation were blocked in

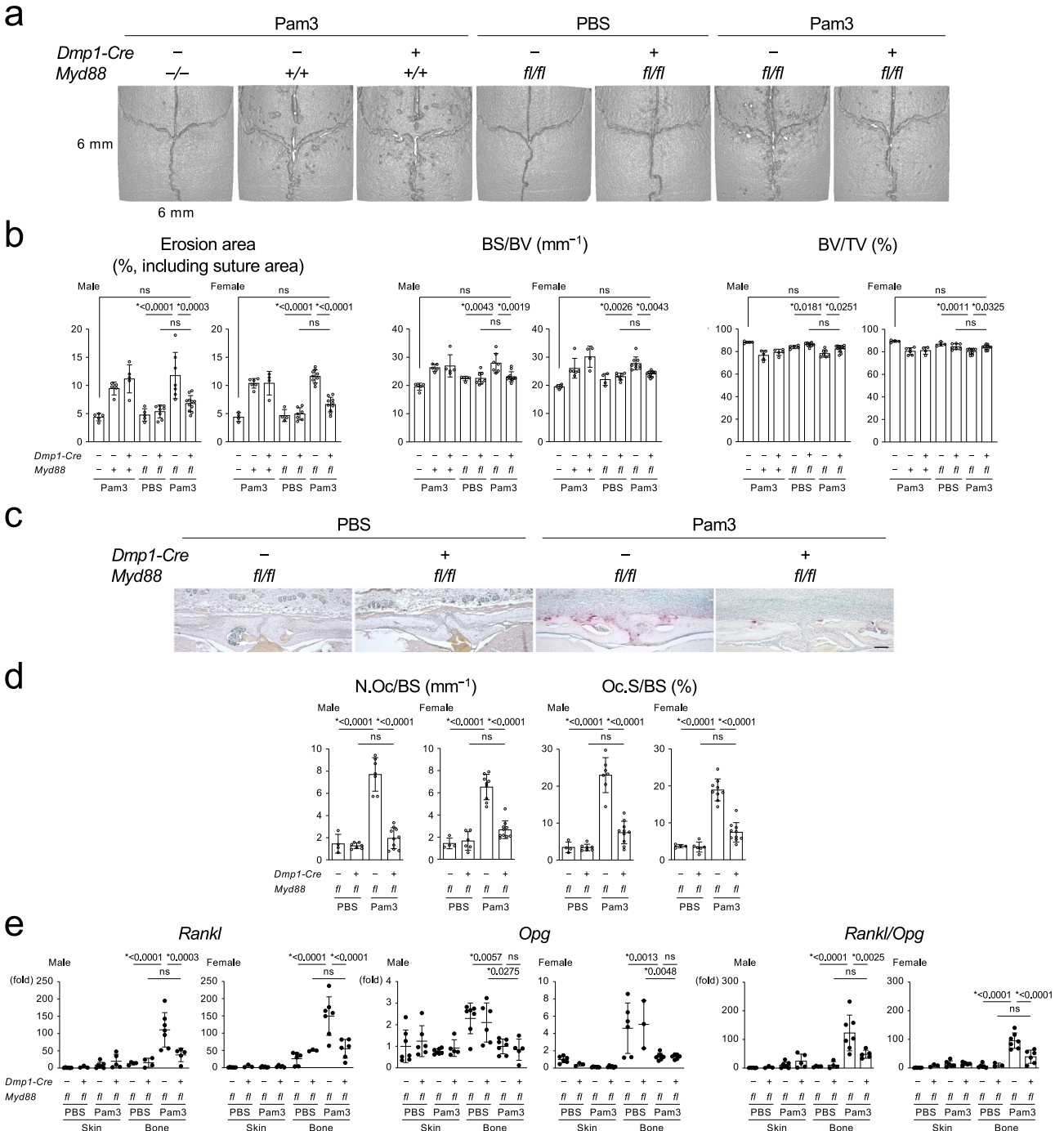

**Fig. 1 | Lack of MYD88 in osteocytes and mature osteoblasts rescues calvarial osteolysis induced by Pam3CSK4 injection. a** MicroCT images of the calvaria. Representative images from each group of male mice ($n \geq 5$/group). **b** Erosion area, BS/BV, and BV/TV of the calvaria ($n = 5, 5, 5, 5, 8, 7, 11$ in male, $n = 4, 6, 4, 4, 7, 9, 10$ in female). **c** TRAP staining of the calvarial bone. Representative images from each group of male mice ($n \geq 4$/group). Scale bar = 100 μm. **d** Histomorphometric analysis of osteoclasts on the calvarial bone surface ($n = 4, 7, 7, 10$ in male, $n = 4, 6, 9, 10$

in female). **e** qPCR analysis of *Rankl* ($n = 6, 3, 7, 5, 4, 4, 7, 5$ in male, $n = 6, 3, 6, 5, 6, 3, 7, 6$ in female), *Opg* ($n = 7, 6, 7, 5, 7, 6, 7, 5$ in male, $n = 6, 3, 7, 6, 6, 3, 7, 6$ in female), and their ratio ($n = 6, 3, 7, 5, 4, 4, 7, 5$ in male, $n = 6, 3, 6, 5, 6, 3, 7, 6$ in female) in skin and calvarial bone tissues. **a–e** Pam3 = Pam3CSK4. **b, d, e** Data are presented as mean ± SD. *$p < 0.05$ with one-way ANOVA with Tukey–Kramer test. ns = not significant. Each data point represents a biologically independent mouse. Source data are provided as a Source Data file.

*Pg*-infected *Dmp1-Cre;Rankl*[fl/fl] mice (Supplementary Fig. 9d, e). The *Dmp1-Cre;Myd88*[fl/fl] mice on the *Rag1*[−/−] background showed alveolar bone protection (Fig. 2f), but lack of IL-1R signaling had no impact on alveolar bone loss (Supplementary Fig. 9f). Together, these results show that activation of the TLR2-MYD88-RANKL axis in osteocytes and mature osteoblasts is responsible for alveolar bone resorption in periodontitis due to *Pg*. Further, the data indicate that osteocytes and

mature osteoblasts are the primary source of RANKL in *Pg*-induced periodontitis and that RANKL from T/B lymphocytes has little effect on osteoclastogenesis in the periodontitis model. Importantly, a *Pg* component was identified in the osteocyte lacunar-canalicular system of alveolar bone (Fig. 2g), suggesting that the TLR2-MYD88 axis in osteocytes is directly activated by *Pg*-derived PAMPs to induce RANKL expression responsible for causing alveolar bone resorption.

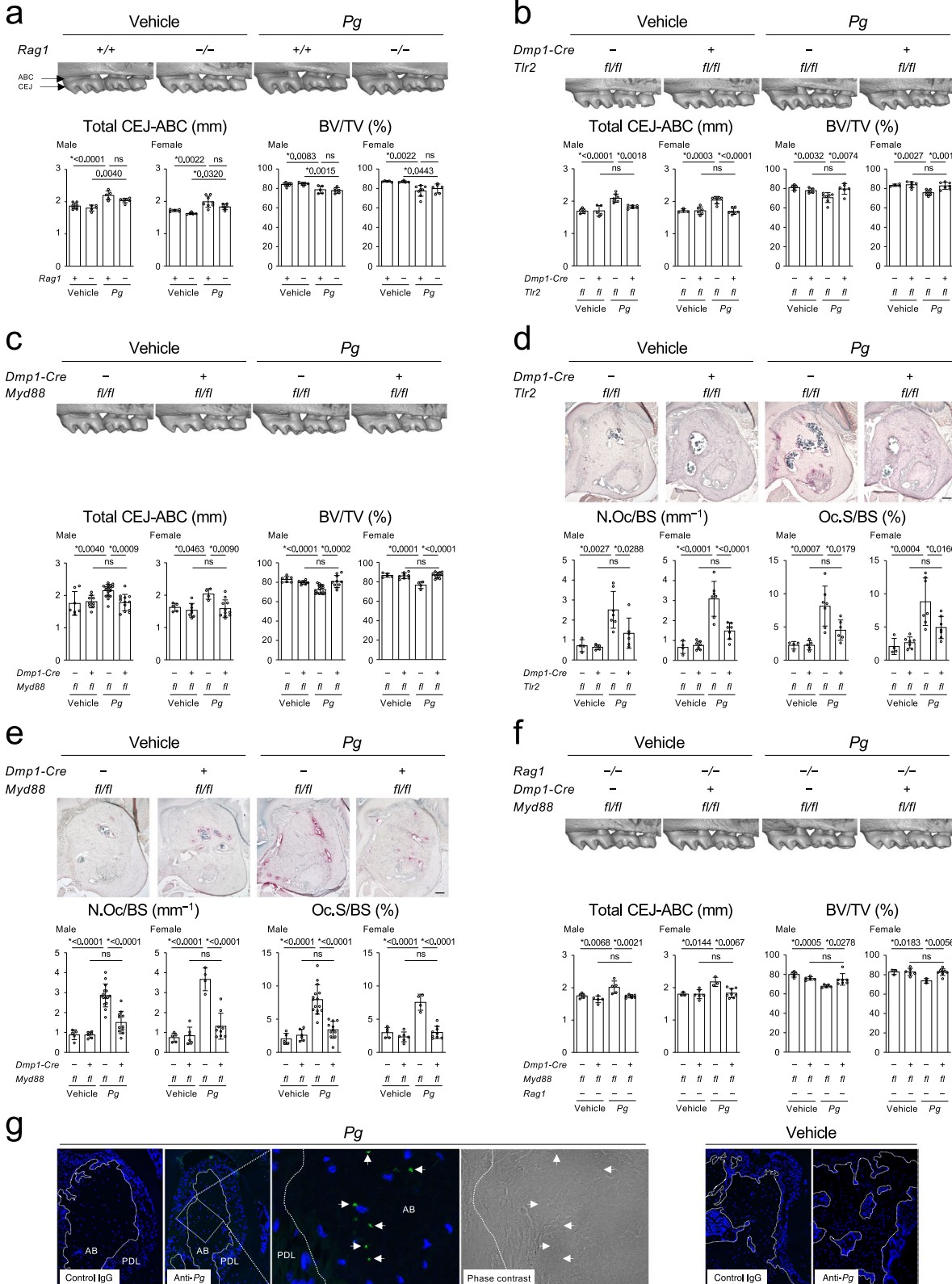

## Targeted restoration of MYD88 in osteocytes and mature osteoblasts is sufficient to cause osteolysis

Osteocytes and osteoblasts are known to express inflammatory cytokines[4,6,17,22–28]. Therefore, we hypothesized that bacterial activation of the MYD88 pathway in these cells alone might be sufficient to cause inflammatory osteolysis. We employed the *Myd88^{lsl/lsl}* mice, where MYD88 becomes functionally active only after *Cre* exposure[29] and

confirmed that *Dmp1-Cre* shows no detectable *Myd88* gene restoration in immune cells and hematopoietic tissues (Supplementary Fig. 10). The *Dmp1-Cre;Myd88^{lsl/lsl}* mice injected with Pam3 or infected with *Pg* showed osteolysis comparable to *Myd88^{+/+}* mice (Fig. 3a, b). In both osteolysis models, osteoclast induction and *Rankl* elevation were observed in bone tissues from *Dmp1-Cre;Myd88^{lsl/lsl}* mice (Fig. 3c, d). H&E staining revealed considerable infiltration of inflammatory cells

**Fig. 2 | The TLR2-MYD88 axis in osteocytes and mature osteoblasts regulates osteolysis in periodontitis induced by Porphyromonas gingivalis infection.** **a**–**c**, **f** Top: MicroCT images of the right maxilla. Representative images from each group of male mice ($n \geq 5$/group). Buccal side view. Bottom: The total CEJ-ABC distance of the right maxillary molars and alveolar BV/TV underneath the second molar of the right maxilla. **a** $n = 8, 5, 5, 6$ in male. $n = 5, 5, 8, 6$ in female. **b** $n = 5, 5, 7, 6$ in male. $n = 4, 6, 7, 7$ in female. **c** CEJ-ABC: $n = 7, 9, 17, 13$ in male. $n = 5, 8, 4, 11$ in female. BV/TV: $n = 7, 8, 16, 10$ in male. $n = 5, 8, 4, 11$ in female. **f** $n = 5, 5, 5, 7$ in male. $n = 3, 6, 3, 8$ in female. **d**, **e** Top: TRAP staining of the alveolar bone. Representative images from each group of male mice ($n \geq 4$/group). Scale bar = 100 μm. Bottom:

Histomorphometric analysis of osteoclasts on the alveolar bone surface. **d** $n = 4, 5, 7, 6$ in male. $n = 4, 7, 7, 7$ in female. **e** $n = 5, 6, 15, 11$ in male. $n = 5, 6, 4, 9$ in female. **g** Immunofluorescence images of the *Pg* component detected in the alveolar bone (green, indicated by arrows). Nuclei: DAPI (blue). AB = Alveolar bone. PDL = Periodontal ligament. Representative images from three independent experiments. Scale bar = 100 μm. **a**–**g** *Pg* = *Porphyromonas gingivalis*. **a**–**f** Data are presented as mean ± SD. *$p < 0.05$ with one-way ANOVA with Tukey–Kramer test. ns = not significant. Each data point represents a biologically independent mouse. Source data are provided as a Source Data file.

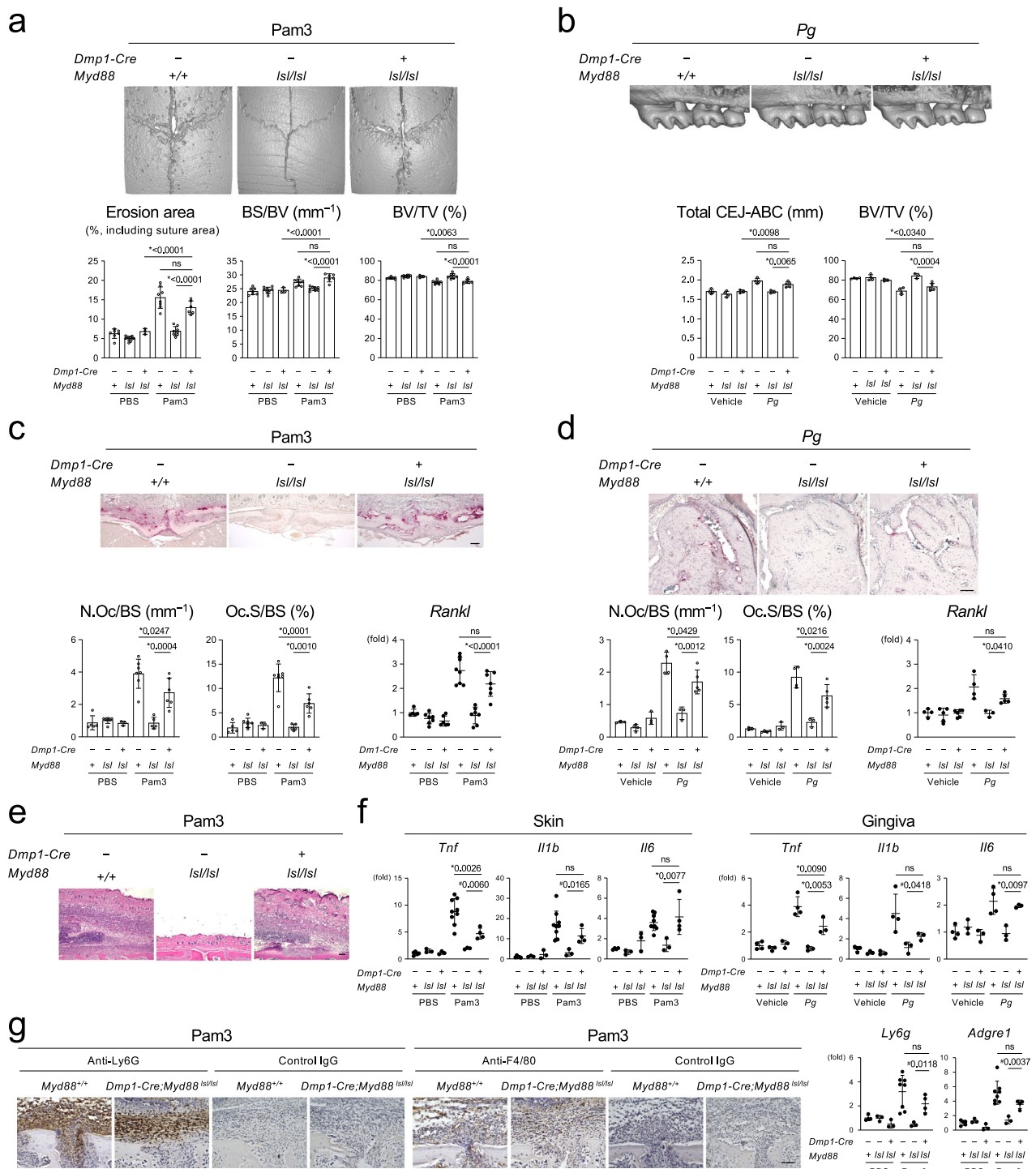

**Fig. 3 | Targeted activation of the MYD88 pathway in osteocytes and mature osteoblasts is sufficient to trigger calvarial osteolysis and alveolar bone loss.** **a** Top: MicroCT images of the calvaria. Representative images from each group of male mice injected with Pam3CSK4 ($n \geq 6$/group). Bottom: Erosion area ($n = 7, 10, 3, 8, 8, 6$), BS/BV ($n = 7, 9, 3, 8, 8, 6$), and BV/TV ($n = 7, 9, 3, 8, 8, 6$) of the calvaria. **b** Top: MicroCT images of the right maxilla. Representative images from each group of male mice inoculated with *Pg* ($n \geq 3$/group). Buccal side view. Bottom: The total CEJ-ABC distance of the right maxillary molars and alveolar BV/TV underneath the second molar of the right maxilla. $n = 4, 4, 3, 4, 3, 5$. **c** Top: TRAP staining of the calvarial bone. Representative images from each group of male mice injected with Pam3CSK4 ($n \geq 5$/group). Scale bar = 100 μm. Bottom left: Histomorphometric analysis of osteoclasts on the calvarial bone surface. $n = 5, 7, 3, 7, 5, 6$. Bottom right: qPCR analysis of *Rankl* in calvarial bone tissue. $n = 5, 7, 6, 8, 7, 7$. **d** Top: TRAP staining of the alveolar bone. Representative images from each group of male mice inoculated with *Pg* ($n \geq 3$/group). Scale bar = 100 μm. Bottom left: Histomorphometric analysis of osteoclasts on the alveolar bone surface. $n = 3, 4, 3, 4, 3, 5$. Bottom right: qPCR analysis of *Rankl* in alveolar bone tissue. $n = 4, 5, 6, 4, 3, 5$. **e** H&E staining of skin tissue on the intersection of the coronal and sagittal sutures. Representative images from each group of male mice injected with Pam3CSK4 ($n \geq 3$/group). Scale bar = 100 μm. **f** qPCR analysis of inflammatory cytokines in skin tissue on the calvaria and gingival tissue. Skin *Tnf* and *Il1b* ($n = 5, 3, 3, 9, 3, 4$). Skin *Il6* ($n = 4, 3, 3, 9, 3, 4$). Gingival *Tnf*, *Il1b*, and *Il6* ($n = 4, 3, 3, 4, 3, 3$). **g** Left: Immunohistochemical staining of neutrophils and macrophages in skin tissue on the calvaria. Representative images from three independent experiments with similar results. Scale bar = 100 μm. Right: qPCR analysis of macrophage and neutrophil marker genes in skin tissue on the calvaria. *Ly6g* ($n = 4, 3, 3, 8, 3, 4$). *Adgre1* ($n = 5, 3, 3, 8, 3, 4$). **a**–**g** Data from male mice. Female mice showed the similar results. Pam3 = Pam3CSK4. *Pg* = *Porphyromonas gingivalis*. **a**–**d**, **f**, **g** Data are presented as mean ± SD. *$p < 0.05$ with one-way ANOVA with Tukey–Kramer test. #$p < 0.05$ when two-tailed unpaired *t*-test was used (**f**, **g**). ns = not significant with ANOVA. Each data point represents a biologically independent mouse. Source data are provided as a Source Data file.

on the calvaria of Pam3-injected *Dmp1-Cre;Myd88^{lsl/lsl}* mice (Fig. 3e). Expression levels of inflammatory cytokines were increased in skin lesions overlying the calvaria of Pam3-injected *Dmp1-Cre;Myd88^{lsl/lsl}* mice and gingiva of *Pg*-infected *Dmp1-Cre;Myd88^{lsl/lsl}* mice compared to *Myd88^{lsl/lsl}* mice treated with Pam3 or *Pg*, respectively (Fig. 3f). Immunohistochemical staining and qPCR analysis of skin lesions revealed that calvarial lesions of Pam3-injected *Dmp1-Cre;Myd88^{lsl/lsl}* mice contain large numbers of immune cells positive for Ly6G or F4/80 (Fig. 3g). These results suggest that specific activation of the MYD88 pathway in osteocytes and mature osteoblasts is sufficient to trigger bone resorption by osteoclasts and progress inflammation on the bone surface by recruiting inflammatory cells, primarily neutrophils and macrophages.

## Osteocytes express significantly greater amounts of RANKL than osteoblasts in response to PAMPs

Both osteocytes and osteoblasts are the cellular sources of RANKL for osteoclastogenesis[30]. To examine whether the RANKL induction capacity of osteocytes is different from that of osteoblasts, the osteocyte-enriched cell population (Ocy) and the osteoblast-enriched cell population (Ob) were differentially isolated from calvariae of 10-week-old mice by serial digestion with collagenase and EDTA[31] followed by the depletion of hematopoietic cells (Supplementary Fig. 11a–c). The percentage of sclerostin-positive cells equivalent to that in osteocytes in vivo[32–34], little contamination of keratocan-positive cells in Ocy, and much higher expression of osteocyte marker genes and sclerostin in Ocy than in Ob showed that osteocytes are highly enriched in Ocy (Supplementary Fig. 11d–f). Likewise, osteoblasts were highly enriched in Ob. Remarkably, while *Dmp1-Cre* has been shown to be active in both osteocytes and osteoblasts in reporter mice[35–38], we found that it depletes MYD88, TLR2, and RANKL and restores MYD88 with significant specificity to Ocy (Supplementary Fig. 11g).

Ocy stimulated with Pam3, *E. coli* LPS, Pam2CSK4, heat-killed *Pg*, or *Pg* culture supernatant, but not flagellin from *Salmonella typhimurium* (FLA-ST) or single-stranded RNA, expressed significantly more RANKL protein or mRNA than Ob, while basal *Rankl* levels were higher in Ocy as reported[9] (Fig. 4a, b and Supplementary Fig. 12a). The *Rankl* induction was faster in Ocy than in Ob. Of note is that stimulation with Pam3, *E. coli* LPS, or Pam2CSK4 provoked biphasic *Rankl* elevation that peaked at 8 and 48 h. Genetic depletion and pharmacological blocking of MYD88 abrogated the RANKL mRNA and protein induction in Ocy (Supplementary Fig. 12b–d). Co-culture of Ocy with bone marrow-derived osteoclast precursors from *Myd88^{-/-}* mice in the presence of Pam3 or *E. coli* LPS caused more significant osteoclast formation than Ob in a RANKL-dependent manner (Fig. 4c–f). Increased expression of osteoclast-associated genes in the Ocy co-culture was suppressed when RANKL was blocked by neutralizing antibodies (Supplementary

Fig. 12e). The soluble form of RANKL was not induced by these stimulations (Supplementary Fig. 12f). These data show that osteocytes have a much greater capacity to support osteoclastogenesis than their precursors, osteoblasts, in the presence of bacterial PAMPs and suggest that activation of the osteocyte MYD88 pathway drives bone resorption via inducing the membrane-bound form of RANKL. Considering the finding that *Dmp1-Cre* deletes and restores MYD88 selectively in Ocy, it is conceivable that the MYD88 pathway in osteocytes rather than that in osteoblasts is playing a major role in regulating osteoclast formation induced by PAMPs in calvarial osteolysis and *Pg*-driven periodontitis models.

## The MYD88-ERK pathway activates CREB and STAT3 to regulate RANKL induction in osteocytes

TLR2 signaling is exclusively mediated by MYD88, while TLR4 signaling is mediated by both MYD88 and toll-like receptor adaptor molecule 1 (TICAM1, also known as TRIF)[39]. To examine the mechanism of how the MYD88 pathway induces RANKL in Ocy, Pam3 was used for Ocy activation. Pam3 stimulation increased the phosphorylation levels of the ERK, JNK, p38, and NF-kB p65 in Ocy (Fig. 5a) by 8 h. Treatment of non-cytotoxic doses of the MEK inhibitor, but not the JNK, p38, or IKK inhibitor, suppressed *Rankl* expression to the basal levels (PBS treatment with no inhibitor) in Pam3-stimulated Ocy in a dose-dependent manner (Fig. 5b), suggesting that the ERK signaling pathway dominantly controls *Rankl* transcription downstream of MYD88 in Ocy. The specific impact of ERK signaling on *Rankl* induction was confirmed in MLO-Y4 cells transfected with MEK1/2 siRNAs (Supplementary Fig. 13a, b).

*Rankl* transcription is governed by the binding of multiple transcription factors (TFs), including CREB, STAT3/5, c-Fos, and Runx2, to the ten regulatory enhancers in stimulation- and cell type-specific fashions[40]. Increased *Rankl* expression was blunted by the CREB and STAT3 inhibitor, but not the STAT5, c-Fos, or Runx2 inhibitor (Fig. 5c), suggesting that CREB and STAT3 are critical TFs for *Rankl* induction downstream of MYD88 and ERK. The critical roles of CREB and STAT3 on *Rankl* regulation was verified in MLO-Y4 cells transfected with CREB or STAT3 siRNAs (Supplementary Fig. 13c, d). Consistent with the results, Pam3 treatment enhanced the phosphorylation levels of CREB and STAT3 (Fig. 5d). Dose-dependent *Rankl* suppression by the CREB inhibitor in the absence of Pam3 implied that constitutive activation of CREB is important to maintain the basal RANKL level in osteocytes. We found that T6167923 or U0126 treatment and genetic ablation of the MYD88 pathway blocked the phosphorylation of CREB and STAT3, confirming that the MYD88-ERK pathway activates CREB and STAT3 in Pam3-stimulated Ocy (Fig. 5e, f, Supplementary Fig. 13e). Finally, CUT & RUN assays showed that Pam3 increased the binding of CREB and STAT3 to the D2/D4/D5/T3 and D2/D5/D6 enhancers of the *Rankl* gene, respectively (Fig. 5g). Similarly, *E. coli* LPS activated the ERK-CREB/

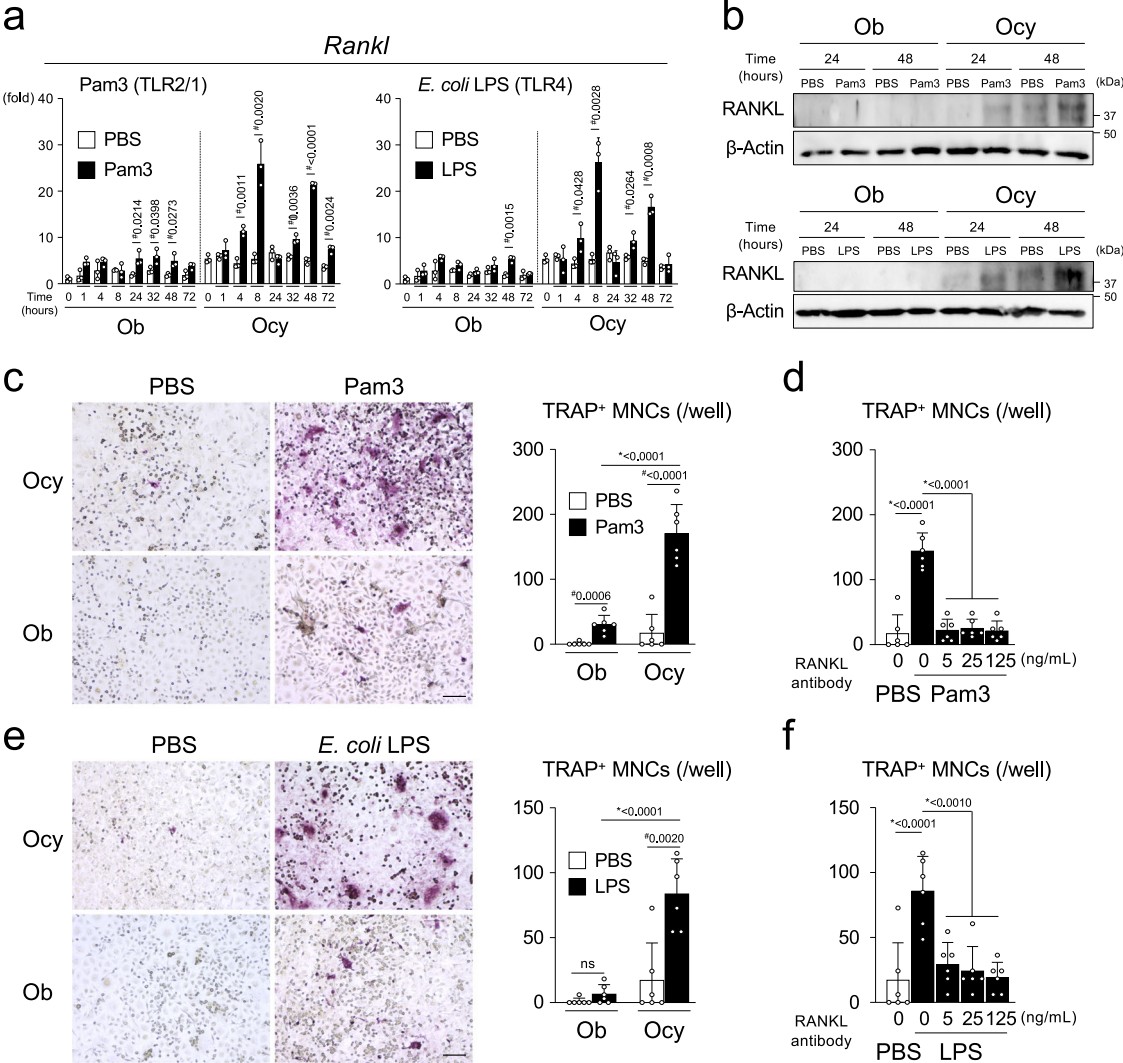

**Fig. 4 | PAMPs induce more robust RANKL expression in the osteocyte-enriched cell population than the osteoblast-enriched cell population. a** qPCR analysis of *Rankl* in the osteoblast-enriched cell population (Ob) and osteocyte-enriched cell population (Ocy) stimulated with Pam3CSK4 (100 ng/mL), *E. coli* LPS (100 ng/mL), or PBS every 24 h. Average expression levels in Ob treated with PBS for 0 h were set as 1. Representative data from five independent experiments with similar results, each with three replicates (*n* = 3). **b** Western blotting of RANKL with cell lysates from Ob and Ocy stimulated with Pam3CSK4, *E. coli* LPS, or PBS. Representative images from five independent experiments with similar results. **c–f** Co-culture of bone marrow-derived M-CSF-dependent macrophages from *Myd88*[−/−] male mice with Ob or Ocy isolated from wild-type (*Myd88*[+/+]) C57BL/6 J male mice. Cells were stimulated with Pam3CSK4, *E. coli* LPS, or PBS for 7 days. **c, e** Formation of TRAP + multinucleated cells (MNCs) and numbers of TRAP + MNCs per well. Representative images and data (*n* = 6/group) from three independent experiments with similar results. Scale bar = 100 μm. **d, f** Numbers of TRAP + MNCs per well in the presence and absence of RANKL neutralizing antibody for 7 days. Representative data from three independent experiments with similar results (*n* = 6/group). **a–d** Pam3 = Pam3CSK4. **a, c–f** Data are presented as mean ± SD. **a** #*p* < 0.05 with two-tailed unpaired *t*-test. **c, e** #*p* < 0.05 with two-tailed unpaired *t*-test. *\**p* < 0.05 when one-way ANOVA with Tukey–Kramer test was used. ns = not significant with *t*-test. **d, f** *\**p* < 0.05 with one-way ANOVA with Tukey–Kramer test. Source data are provided as a Source Data file.

STAT3 pathways to increase *Rankl* expression (Supplementary Fig. 14a–d). Together, these results indicate that CREB and STAT3 activation mediated by the MYD88-ERK pathway regulates early *Rankl* induction by 8 h in Ocy stimulated with Pam3.

**FBXL19 and PDLIM2 regulate RANKL expression via CREB/CBP and STAT3 ubiquitination in osteocytes**
Next, we sought to identify the mechanism by which Pam3 increases *Rankl* transcription at later time points after 8 h. We discovered that blocking the MYD88 pathway by T6167923 significantly decreased CREB, CBP, and STAT3 proteins, but not mRNAs, in Ocy after 12 h (Fig. 6a, b). Similarly, genetic deletion of MYD88 suppressed protein levels of these TFs, but not mRNA, in Ocy (Fig. 6c, d). Restoration of CREB, CBP, and STAT3 protein levels in the presence of MG132, but not

leupeptin, suggested that destabilization of these TFs is mediated by the ubiquitin-proteasome pathway (Fig. 6e, f).

Oppositely, Pam3 activation increased the CREB, CBP, and STAT3 protein stability with no changes in their mRNA levels (Fig. 7a and Supplementary Fig. 15). Decreased lysine (K) 48-ubiquitination of CREB, CBP, and STAT3 after Pam3 stimulation confirmed that the ubiquitin-proteasome pathway controls the stability of these TFs downstream of MYD88 (Fig. 7b–d). Further, we investigated the ubiquitin ligases of CREB, CBP, and STAT3 in Ocy. F-box and leucine-rich repeat protein 19 (FBXL19) is an E3 ubiquitin ligase that ubiquitinates the CBP[41]. We found that FBXL19 protein, not mRNA, was destabilized in Ocy after stimulation of the MYD88 pathway (Fig. 7e and Supplementary Fig. 16a). Therefore, the impact of Pam3 on the formation of molecular complexes containing FBXL19 and CREB/

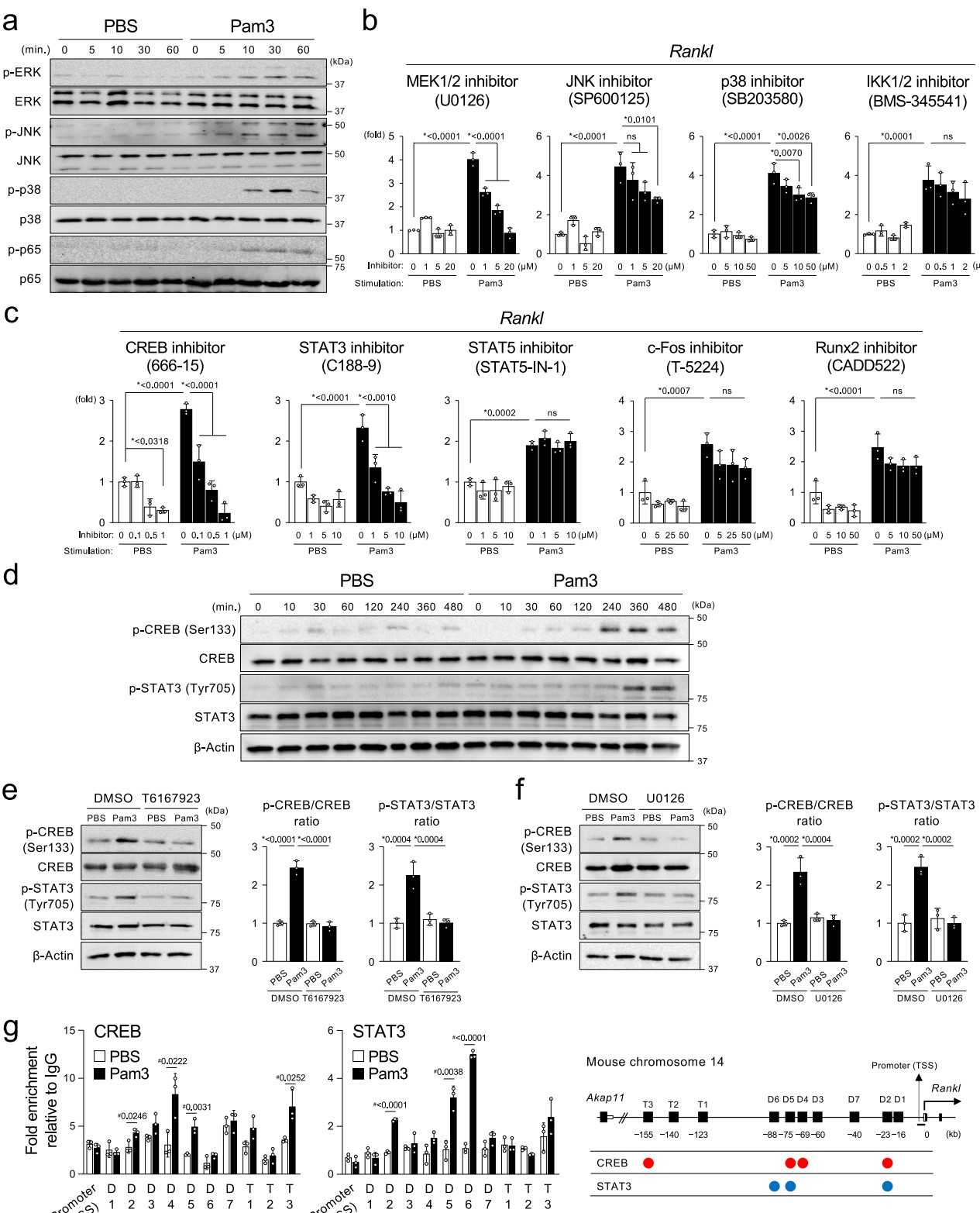

CBP was examined. CREB and CBP formed molecular complexes with FBXL19, and Pam3 stimulation decreased the complex formation (Fig. 7f). Lentiviral overexpression of FBXL19 in osteocytic IDG-SW3 cells increased K48-ubiquitination of CREB and CBP and decreased these protein levels (Fig. 7g). Indeed, FBXL19 overexpression suppressed RANKL protein and mRNA levels in osteocytic IDG-SW3 cells (Fig. 7h, i). Oppositely, knocking down *Fbxl19* increased CREB and CBP proteins and upregulated RANKL protein

and mRNA with decreases in the ubiquitination of CREB and CBP (Supplementary Fig. 16b–e).

PDLIM2 is an E3 ubiquitin ligase that regulates STAT3 degradation[42]. We found that stimulation of the MYD88 pathway destabilizes PDLIM2 protein and mRNA in Ocy (Fig. 7j and Supplementary Fig. 17a). STAT3 formed complexes with PDLIM2, and Pam3-stimulation decreased the complex formation in Ocy (Fig. 7k). Lentiviral overexpression of PDLIM2 increased K48-ubiquitination of

**Fig. 5 | Activation of the MYD88 pathway induces RANKL via the ERK-CREB/ STAT3 axes in the osteocyte-enriched cell population. a** Western blotting of phosphorylated (p) and total ERK, JNK, p38, and NF-kB p65 proteins in Ocy stimulated with Pam3CSK4 or PBS. min. = minutes. **b, c** qPCR analysis of *Rankl*. Ocy was pretreated with increasing concentrations of inhibitors or vehicle (DMSO) for 2 h before stimulation with Pam3CSK4 or PBS for 8 h. Representative data from three independent experiments with similar results, each with three replicates (*n* = 3). **d** Western blotting of phosphorylated (p) and total CREB and STAT3 proteins in Ocy stimulated with Pam3CSK4 or PBS. min. = minutes. **e** Left: Western blotting of phosphorylated (p) and total CREB and STAT3 proteins. Ocy was pretreated with the MYD88 inhibitor T6167923 (20 μM) or vehicle (DMSO) for 2 h before stimulation with Pam3CSK4 or PBS for 8 h. Right: Densitometric analysis of the p-CREB/CREB and p-STAT3/STAT3 ratio using ImageJ. **f** Left: Western blotting of phosphorylated (p) and total CREB and STAT3 proteins. Ocy was pretreated with the MEK1/2 inhibitor U0126 (20 μM) or vehicle (DMSO) for 2 h before stimulation

with Pam3CSK4 or PBS for 8 h. Right: Densitometric analysis of the p-CREB/CREB and p-STAT3/STAT3 ratio using ImageJ. **e, f** Graphs were created from the data of three independent experiments (*n* = 3). **g** CUT & RUN assays of CREB and STAT3 in Ocy stimulated with Pam3CSK4 or PBS for 8 h. Inductions of CREB and STAT3 binding to the *Rankl* promoter and enhancers were quantitated by qPCR relative to the isotype control IgGs (fold enrichment). Representative data from three independent experiments with similar results, each with three replicates (*n* = 3). The diagram shows the mouse *Rankl* promoter and enhancer loci and the binding sites of CREB and STAT3 in Ocy stimulated with Pam3CSK4. TSS = Transcription start site. **a–g** Pam3 = Pam3CSK4. **a, d, e, f** Representative images from more than three independent experiments. **b, c, e, f, g** Data are presented as mean ± SD. **b, c, e, f** \**p* < 0.05 with one-way ANOVA with Tukey–Kramer test. ns = not significant. **g** #*p* < 0.05 with two-tailed unpaired *t*-test. Source data are provided as a Source Data file.

STAT3 and decreased STAT3 protein level in osteocytic IDG-SW3 cells (Fig. 7l). Protein and mRNA levels of RANKL were decreased in osteocytic IDG-SW3 cells overexpressing PDLIM2 (Fig. 7m, n). In contrast, *Pdlim2* knockdown increased STAT3 protein and elevated RANKL protein and mRNA, while it reduced the ubiquitination of STAT3 (Supplementary Fig. 17b–e).

Parathyroid hormone (PTH) stimulates the protein kinase A (PKA) in osteocytes to induce CREB-mediated *Rankl* expression[43,44]. However, PKA inhibition did not suppress *Rankl* induction by Pam3 (Supplementary Fig. 18a). Further, Pam3 stimulation did not increase the binding of CRTC2, a co-activator of CREB critical for PTH-induced RANKL expression[43], to *Rankl* enhancers, suggesting that the mechanism of *Rankl* induction by Pam3 is different from that by PTH (Supplementary Fig. 18b). It is known that IL-6 and TNF-α increase RANKL expression in osteocytes[23,45,46]. Neutralization of IL-6, but not TNF-α, partially suppressed *Rankl* induction by Pam3 (Supplementary Fig. 18c), suggesting that PAMPs-driven *Rankl* up-regulation in osteocytes is primarily regulated by direct TLR activation, but osteocyte-derived cytokines potentiate it in an autocrine or paracrine manner. Thus, the results show that K48-ubiquitination of CREB, CBP, and STAT3 is another key mechanism for inducing RANKL expression downstream of MYD88 in Ocy, where FBXL19 and PDLIM2 play pivotal roles in the ubiquitination process.

### An MYD88 inhibitor protects against alveolar bone loss in periodontitis

We tested the therapeutic impact of an MYD88-specific inhibitor on periodontitis. We discovered that systemic administration of T6167923[47,48] protected against the increase of the CEJ-ABC distance and the decrease of BV/TV of alveolar bone in wild-type mice infected with *Pg* (Fig. 8a, b). These protective effects were accompanied by decreased osteoclast formation on the surface of alveolar bone and reduced expression of osteoclast-associated genes in the alveolar bone (Fig. 8c, d). The data show that pharmacological inhibition of MYD88 suppresses osteoclast induction and alveolar bone resorption caused by periodontitis in vivo, thereby suggesting that MYD88 can be a therapeutic target for bone loss due to bacterial infections, including periodontitis.

## Discussion

The mechanism responsible for osteolysis associated with bone infection has been unclear. Moreover, it remains unknown whether and how the MYD88-mediated host defense system of osteocytes triggered by bacterial infection impacts the skeleton. The current study demonstrates that osteocytes dominantly control bacterially-induced osteolysis by directly integrating the MYD88 pathway activation into the mechanism of RANKL induction responsible for osteoclastogenesis. The pattern recognition receptors TLR2 and TLR4 play a central role in the osteocyte recognition of bacterial

PAMPs. Through these receptors, the MYD88 pathway converts infection signaling into bone-resorption signaling. Our data also show that the impact of the osteocyte MYD88 pathway on bone resorption emerges in pathological settings rather than physiological settings. We established an improved method to separately isolate osteocyte-enriched and osteoblast-enriched cell populations from the calvariae of adult mice. Using this method, we discovered that osteocytes express significantly higher amounts of RANKL than osteoblasts when stimulated with PAMPs, and the 10 kb *Dmp1* promoter is highly specific for osteocytes compared to osteoblasts. Therefore, rescues from calvarial osteolysis and alveolar bone loss in *Dmp1-Cre;Myd88fl/fl* mice were interpreted to demonstrate that the TLR-MYD88 signaling axis in osteocytes dominantly regulates osteolysis in bone infections in the oral and craniofacial regions. Further, the *Dmp1-Cre* specificity to osteocytes showed that osteocytes are the primary source of RANKL in the mechanism of bacterially-induced bone resorption.

RANKL derived from osteocytes has been shown to play a critical role in physiological and pathological bone resorption[8–10,49–53]. But its role in bone infection is poorly understood. In addition, while bacterial PAMPs have been shown to increase RANKL expression in mouse neonatal calvarial osteoblasts and the osteocytic MLO-Y4 cell line in vitro[14,17], these studies have never compared the potential of RANKL induction between osteocytes and osteoblasts directly. We discovered that activation of the MYD88 pathway promotes the expression of RANKL mRNA and protein much more robustly in the osteocyte-enriched cell population than in the osteoblast-enriched cell population. Considering the fact that osteocytes are the most abundant cells in the bone, the much higher RANKL induction capacity of osteocyte-enriched cells can be the rationale behind our conclusion that osteocytes play a dominant role in osteoclast formation in bacterial bone infection.

Osteocytes regulate hematopoiesis and secrete a wide range of inflammatory factors when stimulated with bacterial pathogens[4,22,54,55]. Nonetheless, the in vivo impact of such osteocyte-derived factors on bone has been undetermined. To our surprise, *Dmp1-Cre;Myd88lsl/lsl* mice revealed that osteocyte-selective restoration of the MYD88 pathway is sufficient to trigger and develop inflammation and osteolysis on the bone surface, suggesting that osteocytes transform into powerful inflammatory cells that provide not only RANKL but also pro-inflammatory "osteocytokines" and "osteochemokines" when stimulated with bacterial PAMPs. Therefore, "osteocyte inflammation" may regulate the migration of immune and osteoclast progenitor cells to the bone surface in bone infectious diseases. Cooperative activation of neutrophils and macrophages is likely to be necessary to boost the development of inflammatory lesions on the bone surface. It would be interesting to determine what osteocyte-derived inflammatory mediators play key roles in recruiting neutrophils and macrophages to the bone surface

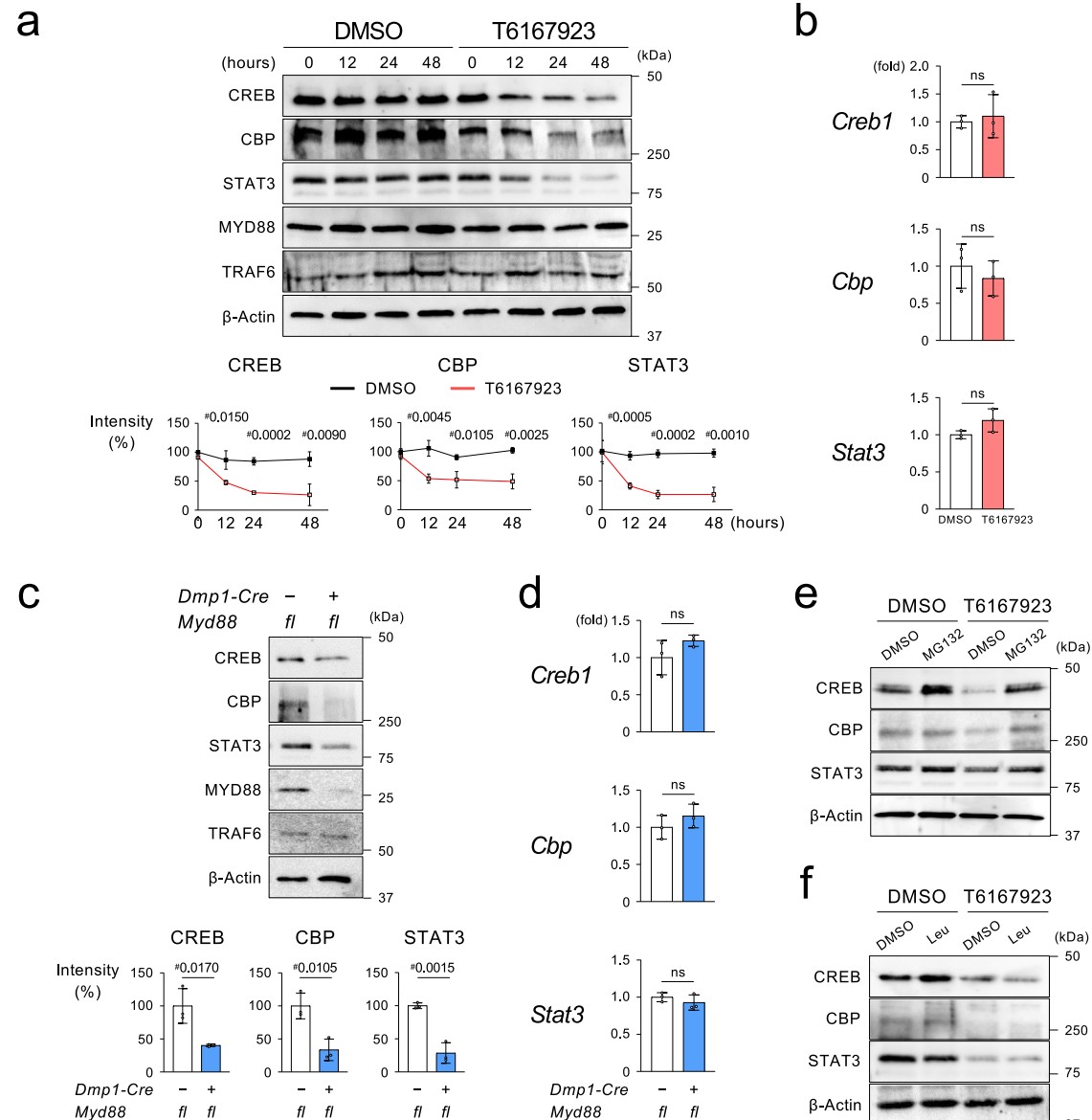

**Fig. 6 | The MYD88 pathway regulates CREB, CBP, and STAT3 protein stability in the osteocyte-enriched cell population. a** Western blotting of CREB, CBP, and STAT3 proteins in Ocy isolated from wild-type ($Myd88^{+/+}$) male mice. Ocy was treated with T6167923 or vehicle (DMSO) every 24 h. Graphs show the intensities of the protein bands relative to those of DMSO-treated Ocy at 0 h (%). Graphs were created from the data of three independent experiments ($n = 3$). Normalized by β-Actin. **b** qPCR analysis of *Creb1*, *Cbp*, and *Stat3* in male $Myd88^{+/+}$ Ocy treated with or without T6167923 (20 μM) for 48 h. T6167923 was added to the cultures every 24 h. Representative data from three independent experiments with similar results, each with three replicates ($n = 3$). **c** Western blotting of the indicated proteins in Ocy isolated from *Dmp1-Cre;Myd88$^{fl/fl}$* and *Myd88$^{fl/fl}$* male mice. Graphs show the intensities of the protein bands from *Dmp1-Cre;Myd88$^{fl/fl}$* mice relative to those from

$Myd88^{fl/fl}$ mice normalized by β-Actin (%). Graphs were created from the data of three independent experiments ($n = 3$). **d** qPCR analysis of *Creb1*, *Cbp*, and *Stat3* in Ocy isolated from *Dmp1-Cre;Myd88$^{fl/fl}$* and *Myd88$^{fl/fl}$* male mice. Representative data from three independent experiments, each with three replicates ($n = 3$). **e,f** Western blotting of CREB, CBP, and STAT3 in Ocy isolated from $Myd88^{+/+}$ male mice. Ocy was treated with a proteasomal inhibitor MG132 (1 μM) or a lysosomal inhibitor leupeptin (100 μM) for 3 h before treatment with T6167923 (20 μM) for 48 h. MG132, leupeptin, and T6167923 were added to the cultures every 24 h. **a, b, c, d** Data are presented as mean ± SD. #$p < 0.05$ with two-tailed unpaired *t*-test. ns = not significant. **a, c, e, f** Representative images from more than three independent experiments. Source data are provided as a Source Data file.

and whether inflamed osteocytes are involved in the mechanism of osteocytic osteolysis. Detailed time-course studies with *Dmp1-Cre;-Myd88$^{fl/fl}$* and *Dmp1-Cre;Myd88$^{lsl/lsl}$* mice will determine the inflammation and bone resorption phases (e.g., initiation, progression, or establishment) critically affected by osteocyte-derived inflammatory mediators, including RANKL. Together, the current study provides genetic evidence that osteocytes are the direct cellular mediators of bone resorption and inflammation. As a result, osteocytes could be the cellular target for blocking the migration of osteoclast progenitor and inflammatory cells to the bone surface.

Mechanistically, activation of the MYD88 pathway increased the phosphorylation levels of CREB and STAT3, and blocking these TFs prevented the *Rankl* induction in osteocytic cells. These data show the critical role of CREB and STAT3 in the *Rankl* regulation of osteocytes. CREB is known to regulate *Rankl* transcription by binding to the D2, D4, and D5 enhancers in the osteoblast-like MC3T3-E1 cells[40]. Our data show that, in the osteocyte-enriched cell population, Pam3 promotes the binding of CREB to the T3 enhancer in addition to the D2, D4, and D5 enhancers. T3 was initially identified in the mechanism of T cell-specific *Rankl* transcription[56]. CREB binding to the T3 is likely

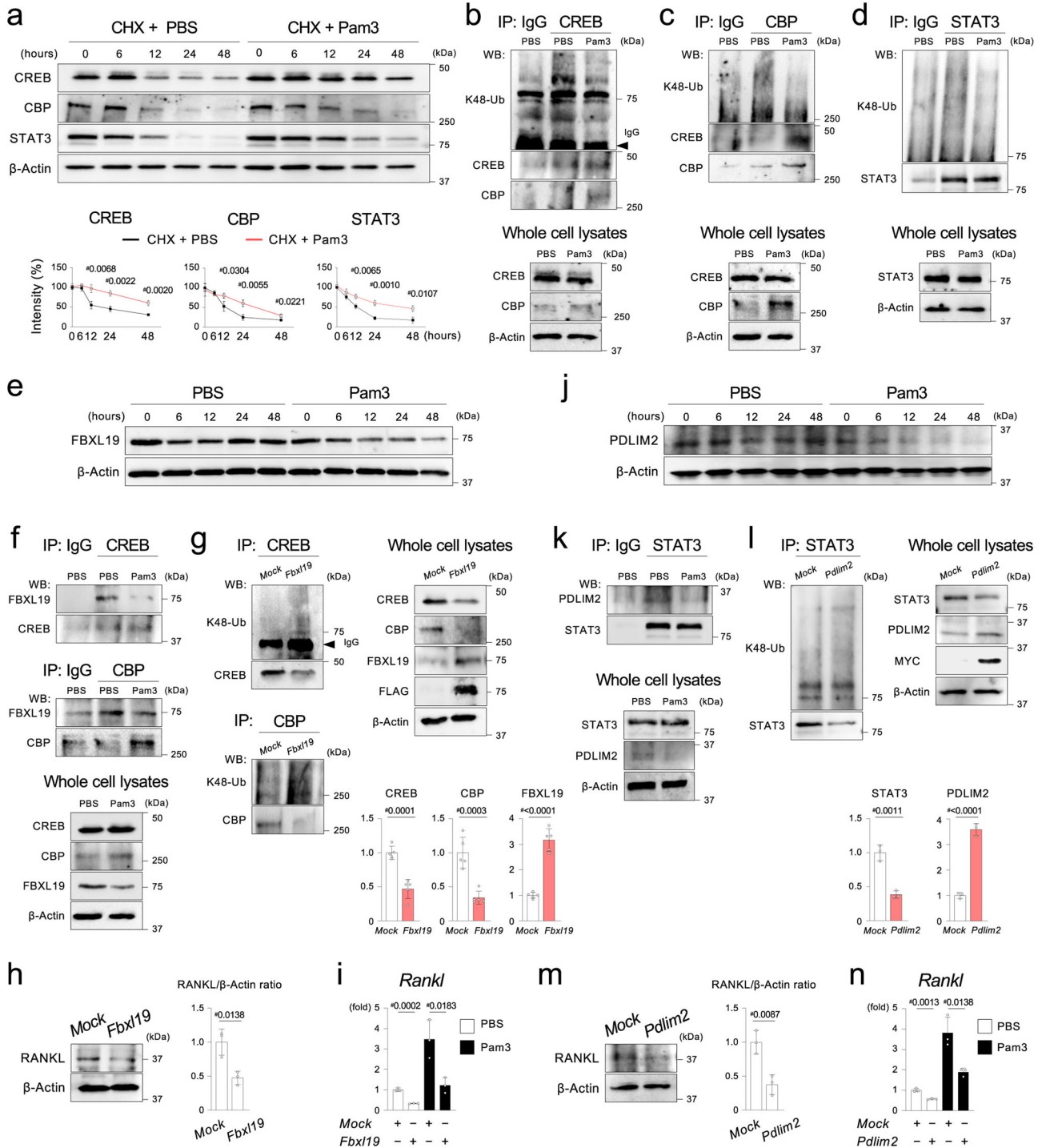

important for the osteocyte-specific mechanism of *Rankl* induction. Similarly, STAT3 plays a regulatory role for *Rankl* transcription by binding to the D4, D5, and D6 enhancers in the ST2 mouse stromal cell line[57,58]. We showed that Pam3 stimulation provokes STAT3 binding to the D2, D5, and D6 enhancers in the osteocyte-enriched cell population. Thus, increased STAT3 binding to the D2 may be involved in the mechanism of more robust *Rankl* induction in osteocytes than osteoblasts. Genome-wide comparison of changes in CREB and STAT3 binding sites in response to Pam3 between osteocytes and other cell types, including osteoblasts and T cells, will identify new osteocyte-specific *Rankl* transcription mechanisms.

The greater *Rankl* induction capacity of osteocyte-enriched cells may also be explained by increased CREB, CBP, and STAT3 protein

stability. We discovered that the MYD88 pathway regulates RANKL expression via the K48-ubiquitination of CREB/CBP and STAT3 and their proteasomal degradation. FBXL19 is an F-box protein that regulates the ubiquitination of CBP, Rac1/3, RhoA, and IL-33R in the Skp1-Cul1-F-box protein (SCF) E3 ubiquitin ligase complex[41,59–63]. Our data showed that CREB is a ubiquitination target of FBXL19 in osteocytes, and FBXL19 regulates osteocyte RANKL production via CREB and CBP. PDLIM2 is an E3 ubiquitin ligase for STAT3 and plays an important role in T cell and macrophage functions[42,64–66]. However, its role in the skeletal system remained unknown. We showed that a critical role of PDLIM2 in the skeletal system is to control RANKL in osteocytes via targeted ubiquitination of STAT3. Activation of the MYD88 pathway reduced the formation of molecular complexes containing CREB/CBP

**Fig. 7 | Activation of the MYD88 pathway induces RANKL by decreasing CREB, CBP, and STAT3 ubiquitination in the osteocyte-enriched cell population.** **a** Degradation kinetics of CREB, CBP, and STAT3 proteins in Ocy stimulated with Pam3CSK4 or PBS. Ocy was treated with cycloheximide (CHX, 10 μM) and Pam3CSK4/PBS every 24 h. Graphs show the intensities of the protein bands relative to those at 0 h (%). Normalized by β-Actin. **b–d** Western blotting of K48-ubiquitinated proteins after immunoprecipitation (IP) of CREB, CBP, and STAT3. Ocy was stimulated with Pam3CSK4 or PBS for 48 h. **e** Western blotting of FBXL19 in Ocy stimulated with Pam3CSK4 or PBS. **f** IP of CREB and CBP followed by Western blotting for FBXL19. 48 h after Pam3CSK4 or PBS stimulation. **g** Western blotting of K48-ubiquitinated proteins after IP of CREB or CBP. Cell lysates of differentiated IDG-SW3 cells overexpressing FLAG-tagged mouse FBXL19 were used. Graphs show the relative intensities of CREB, CBP, and FBXL19 protein bands against β-Actin in whole cell lysates. **h** Left: Western blotting of RANKL using cell lysates from differentiated IDG-SW3 cells overexpressing FLAG-tagged mouse FBXL19. Right: Relative RANKL protein levels normalized by β-Actin. **i** qPCR analysis of *Rankl* in differentiated IDG-SW3 cells overexpressing FLAG-tagged mouse

FBXL19 stimulated with Pam3CSK4 or PBS for 48 h. **j** Western blotting of PDLIM2 in Ocy stimulated with Pam3CSK4 or PBS. **k** IP of STAT3 followed by Western blotting for PDLIM2. 48 h after Pam3CSK4 or PBS stimulation. **l** Western blotting of K48-ubiquitinated proteins after IP of STAT3. Cell lysates of differentiated IDG-SW3 cells overexpressing MYC-tagged mouse PDLIM2 were used. Graphs show the relative intensities of STAT3 and PDLIM2 protein bands against β-Actin in whole cell lysates. **m** Left: Western blotting of RANKL using cell lysates from differentiated IDG-SW3 cells overexpressing MYC-tagged mouse PDLIM2. Right: Relative RANKL protein levels normalized by β-Actin. **n** qPCR analysis of *Rankl* in differentiated IDG-SW3 cells overexpressing MYC-tagged mouse FBXL19 stimulated with Pam3CSK4 or PBS for 48 h. **a–f, i–k, n** Pam3 = Pam3CSK4. **a, g, h, i, l, m, n** Graphs were created from the data of three independent experiments (*n* = 3). **i, n** Representative data from three independent experiments with similar results, each with three replicates (*n* = 3). **a, g, h, i, l, m, n** Data are presented as mean ± SD. #*p* < 0.05 with two-tailed unpaired *t*-test. **a–h, j–m** Representative images from more than three independent experiments. Source data are provided as a Source Data file.

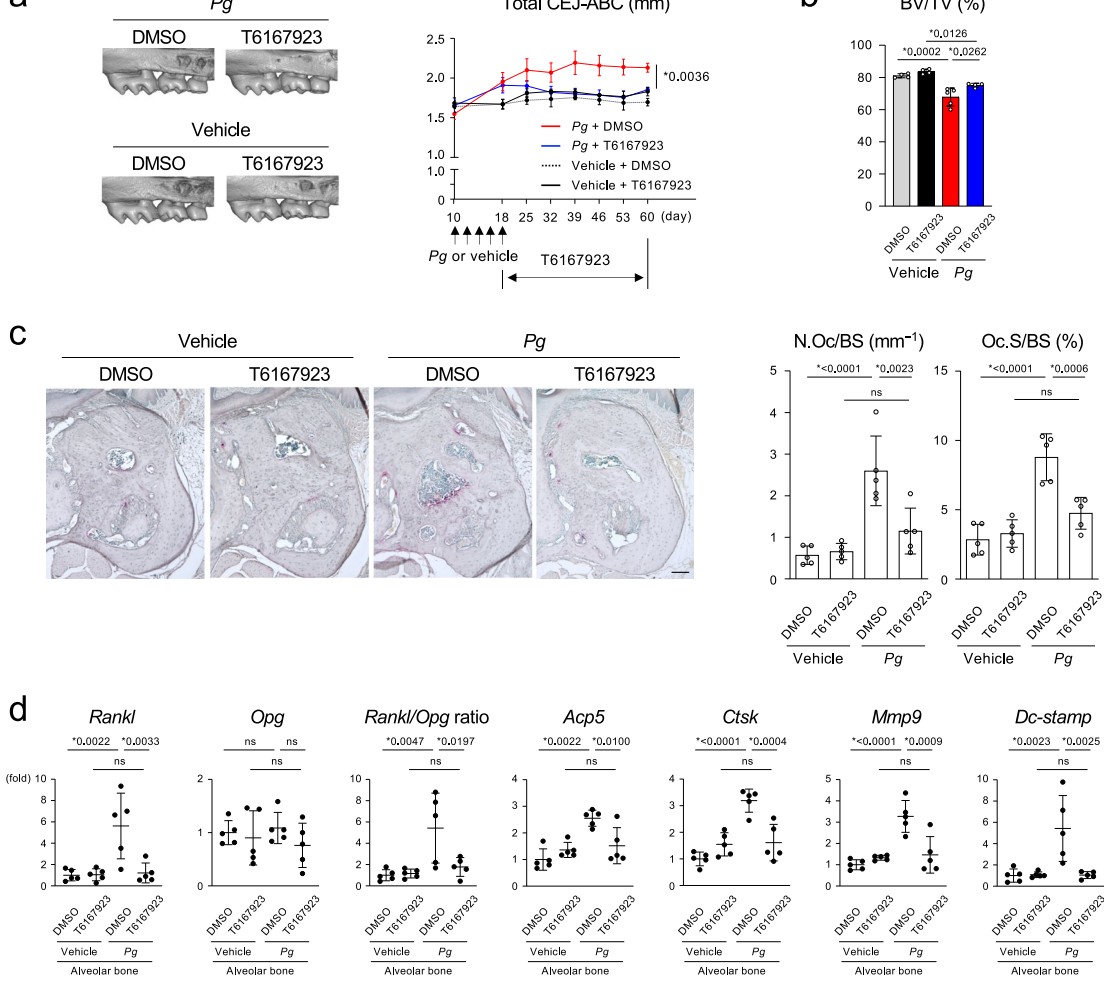

**Fig. 8 | Administration of an MYD88 inhibitor protects against alveolar bone loss in periodontitis induced by Porphyromonas gingivalis infection. a** Left: MicroCT images of the right maxilla from *Porphyromonas gingivalis* (*Pg*)- or vehicle-inoculated wild-type C57BL/6 J male mice administered with T6167923 or DMSO. Buccal side view. Representative images from each group of male mice. Right: The total CEJ-ABC distance at the right maxillary molars. **b** Alveolar BV/TV underneath the right maxillary second molar. **a, b** *Pg* + DMSO (*n* = 5), *Pg* + T6167923 (*n* = 4), Vehicle + DMSO (*n* = 4), Vehicle + T6167923 (*n* = 4). **c** Left: TRAP staining of the

alveolar bone. Representative images from each group of male mice (*n* = 5/group). Scale bar = 100 μm. Right: Histomorphometric analysis of osteoclasts on the alveolar bone surface. **d** qPCR analysis of osteoclast-associated genes in the alveolar bone (*n* = 5/group). **a–d** Data are presented as mean ± SD. *p* < 0.05 with one-way ANOVA with Tukey–Kramer test. ns = not significant. Each data point represents a biologically independent mouse. Source data are provided as a Source Data file.

and FBXL19 in osteocyte-enriched cells while reducing the ubiquitination of CREB/CBP. Similarly, it decreased the interaction of PDLIM2 with STAT3 with decreased STAT3 ubiquitination. These results suggest that reduced interactions of CREB/CBP and STAT3 to their ubiquitin ligases are a primary cause for promoting CREB/CBP and STAT3 protein stabilities responsible for RANKL induction in bacterially-inflamed osteocytes. The down-regulation of FBXL19 and PDLIM2 protein levels following the MYD88 pathway activation may be an underlying mechanism for decreasing these molecular interactions. Because overexpression of FBXL19 or PDLIM2 suppresses RANKL in osteocytic cells, preventing the reduction of FBXL19 and PDLIM2 proteins or increasing the stability of these ubiquitin ligases will be a strategy for suppressing RANKL in osteocytes. FBXL19 degradation is regulated by FBXW17-mediated ubiquitination and CBP-mediated acetylation[67,68]. Manipulating these mechanisms of FBXL19 degradation may be an approach to suppress osteoclastogenesis in bone infection. Regulatory mechanisms of PDLIM2 degradation and mRNA transcription need to be revealed to identify strategies for suppressing RANKL via PDLIM2. As MYD88 signaling activates the NF-kB pathway and NF-kB p65 is another ubiquitination target of PDLIM2[65], it might be possible that PDLIM2-mediated p65 ubiquitination is also decreased in Pam3-stimulated osteocytes to upregulate pro-inflammatory gene expression. Together, our data show that FBXL19 and PDLIM2 regulate RANKL induction downstream of MYD88 in osteocytes, suggesting that these ubiquitin ligases are therapeutic targets for osteolysis in bone infection.

Notably, activation of the MYD88 pathway with Pam3 increased the stabilization of CREB, CBP, and STAT3 proteins after 12 h in the osteocyte-enriched cell population. Because *Rankl* expression peaks at 8 and 48 h after Pam3 stimulation, increased stability of CREB, CBP, and STAT3 is likely to be involved in the *Rankl* induction at the late phase. In contrast, increased phosphorylation of CREB and STAT3 by 8 h, which leads to their binding to *Rankl* enhancers, is likely responsible for the *Rankl* induction at the early phase. Paracrine or autocrine effects of osteocyte-derived inflammatory cytokines (e.g., IL-6) on the stability of these TFs need to be investigated to determine if signaling pathways other than the MYD88 pathway regulate the degradation of CREB/CBP and STAT3 through FBXL19 and PDLIM2, respectively. Our data showed that single inhibition of CREB or STAT3 by inhibitor or siRNA treatment efficiently suppressed Pam3-induced *Rankl* expression, suggesting that CREB and STAT3 cooperatively regulate RANKL transcription in osteocytes. Intriguingly, Pam3 injection decreased *Opg* levels in bone regardless of the MYD88 deletion, indicating that MYD88-independent mechanisms of OPG suppression, most likely in osteoblasts[69,70], contribute to promoting osteolysis in cooperation with osteocyte RANKL when bones are infected. Ultimately, these molecular mechanisms of inducing RANKL expression identified by using Pam3 will need to be validated in osteocytes stimulated with live bacteria.

In *Pg*-associated periodontitis, T and B cells are potential sources of RANKL[19,71]. However, no significant rescue from the alveolar bone loss was detected in *Pg*-infected RAG1-deficient mice that lack both T and B cells, suggesting that RANKL from these lymphocytes is not critically important for osteoclast formation in the *Pg*-induced periodontitis model. In contrast, both *Rag1*[−/−]; *Dmp1-Cre*; *Myd88*[fl/fl] and *Dmp1-Cre;Rankl*[fl/fl] mice exhibited a significant rescue from alveolar bone osteolysis, thereby confirming that osteocytes are the primary source of RANKL in *Pg*-induced periodontitis. These findings suggest that regardless of the immune cell types involved in the alveolar bone loss, osteocytes are the predominant and direct source of RANKL responsible for osteoclast formation driven by *Pg* infection. Importantly, our data from RAG1-deficient mice indicate that RANKL induction in osteocytes is independent of T and B cell activation in *Pg* periodontitis. It would be of interest to investigate to what extent osteocyte RANKL is required for osteoclastogenesis in the ligature-induced periodontitis model where IL-17 from $T_H17$ cells is critically inducing RANKL expression on periodontal ligament cells and osteoblasts[72].

Previously, *Pg* was shown to invade the epithelial and periodontal ligament cells to stimulate inflammatory responses leading to periodontitis in mice and humans[73–75]. Our study showed that the osteocyte lacunar-canalicular system could harbor *Pg* components, and heat-killed *Pg* and *Pg* culture supernatant increase *Rankl* expression in osteocyte-enriched cells. These results indicate that *Pg*-derived pathogens interact directly with osteocytes to stimulate *Rankl* expression via the TLR2-MYD88 pathway in vivo. Pathogenic components of *Pg* could reach the osteocytes through the local blood circulation system in the oral cavity. Thus, it will be interesting to examine if live *Pg* invades and survives in the lacunar-canalicular space of the alveolar bone. As *Dmp1-Cre;Myd88*[fl/fl] mice infected with *Pg* show a rescue from alveolar bone loss and *Dmp1-Cre;Myd88*[lsl/lsl] mice infected with *Pg* exhibit alveolar bone loss, we propose that direct activation of the alveolar bone osteocytes by bacterial PAMPs is a pathological mechanism of periodontal bone loss and the MYD88 pathway would be a treatment and prevention target for periodontal diseases. Indeed, in this study, MYD88 inhibitor treatment stopped the progression of alveolar bone loss induced by *Pg*. Ultimately, osteocyte-specific drug delivery systems would need to be developed because systemic MYD88 inhibition will increase the host's susceptibility to bacterial infection and impair bacterial clearance by the host immune system[76–78].

While we have shown the critical roles of the osteocyte MYD88 pathway in bone infection, there are limitations in the current study. Recent reports indicated that *Dmp1-Cre* is active in both osteocytes and mature osteoblasts in Ai9/14 reporter mice[35–38]. However, previous studies have not directly compared the *Dmp1-Cre*-mediated recombination efficiency of targeted genes between primary osteocytes and osteoblasts. We found that the *Dmp1* promoter is highly specific to osteocytes by directly comparing the deletion/restoration efficiency of MYD88, RANKL, and TLR2 proteins between Ocy and Ob. This discrepancy could be due to the difference in assay sensitivity. The Ai9/14 mice may be extremely sensitive to trace amounts of *Cre* expression that are insufficient to delete other floxed transgenes[79]. Alternatively, *Dmp1-Cre* could exhibit the osteocyte specificity in a targeting construct-, target gene-, bone type and location-, or ossification type-dependent manner[79]. Given the marked selectivity of *Dmp1-Cre* for calvarial osteocytes, *Dmp1-Cre* is the most reliable *Cre* driver for current osteocyte studies involving osteoclasts and immune cells since, unlike *Sost-Cre*[10], it shows little or no off-target activity to hematopoietic progenitors differentiating into osteoclasts, T/B lymphocytes, or myeloid cells. In the current study, 1) the greater RANKL induction capacity of Ocy in response to bacterial PAMPs, which can be recapitulated in IDG-SW3 cells (Supplementary Fig. 19), 2) the greater *Dmp1-Cre* specificity to Ocy, and 3) the fact that osteocytes are the most abundant cell type in bone may serve as rationales for concluding that it is the osteocytes that play critical roles in regulating osteolysis in bone infection. However, our method for separating osteocytes from osteoblasts is still suboptimal and imperfect, and Ocy and Ob populations isolated from the calvarial bone are obviously heterogeneous. Therefore, our results can be interpreted to conclude that the MYD88 pathway in both osteocytes and mature osteoblasts, but dominantly the pathway in osteocytes, is playing essential roles in the mechanism of PAMPs-driven osteolysis in oral and craniofacial regions. This interpretation is supported by the data that *Osteocalcin-Cre;Myd88*[fl/fl] mice lacking MYD88 in both Ocy and Ob exhibit the suppression of osteolysis comparable to that in *Dmp1-Cre;Myd88*[fl/fl] mice (Supplementary Fig. 20). Future development of the definitive osteocyte-specific *Cre* mice will be needed. It is also conceivable that the extent to which the osteocyte TLR-MYD88-RANKL axis affects osteolysis depends on the pathogen and the site of bone infection. Our study will

not exclude the possible involvement of T/B cells or other cellular RANKL sources, e.g., MALPs[80], in the mechanism of bone resorption triggered by bacteria. Importantly, the possible MYD88 deletion/restoration in a subset of CAR cells, skeletal muscle cells, and trans-cortical perivascular cells by *Dmp1-Cre* needs to be considered in the interpretation of our findings[35–37,81]. Particularly, it would be of interest to examine if CAR cells are a critical RANKL supplier for osteoclast induction in bone infection.

In summary, our study identified a role of the MYD88 pathway in the skeletal system and revealed that osteocytes are bacteria sensors embedded in the bone that transform into inflammatory cells producing RANKL. We also discovered that the bacterially-stimulated osteocytes, and likely also the bacterially-stimulated mature osteoblasts to some extent, have the capacity to trigger and develop osteolysis directly and independently through RANKL induction and inflammatory cell recruitment (Supplementary Fig. 21). These insights into osteocytes establish the osteocyte as the central regulator of osteoclastogenesis and bone resorption via RANKL. Since activation of the osteocyte MYD88 pathway is a significant cause of inflammatory osteolysis due to bacterial infection, targeting the MYD88 signaling and downstream RANKL regulatory mechanisms in osteocytes would be a treatment strategy for bone destruction in periodontitis and osteomyelitis. Similarly, arthritis patients whose bone destruction is caused by the gain-of-function MYD88 mutation[82] might benefit from the same therapeutic strategy.

## Methods

### Study approval

All mutant mouse lines and experimental procedures were approved by the Institutional Animal Care and Use Committee (IACUC) of the Indiana University School of Medicine and Institutional Biosafety Committee (IBC) of the Indiana University.

### Mice

C57BL/6 J, *Myd88^fl/fl^*, *Myd88^lsl/lsl^*, *Rankl^fl/fl^*, *Il1r1^fl/fl^*, *Rag1^−/−^*, *Myd88^−/−^*, *Tlr2^−/−^*, and *Tlr4^lps-de/lps-del^*, *Dmp1-Cre*, *Osteocalcin-Cre* mice were obtained from the Jackson Laboratory (Bar Harbor, ME, USA). *Tlr2^fl/fl^* mice were provided by Drs. Rojas, Harding, and Boom. All mice were bred and housed under specific-pathogen-free (SPF) conditions except *Myd88^−/−^* and *Myd88^lsl/lsl^* mice that were bred and housed under SPF conditions with autoclaved feed and acidified water (pH 2.3 to 2.8).

### *Cre* recombination analysis of *Myd88* by genomic DNA PCR

Genomic DNA was isolated using the DNeasy Blood and Tissue Kit (Qiagen). Ten ng of DNA was used for PCR reactions.

### Calvarial injection of Pam3CSK4, *E. coli* LPS, *S. aureus* LTA, and *Pg* LPS

Pam3CSK4, *S. aureus* LTA, and *Pg* LPS (100 μg in 20 μL PBS/mouse) were injected onto the calvaria of 10 to 11-week-old mice every other day for 3 times (the first day of injection = day 1). Gene expression was examined by qPCR at day 6. Bones were examined by microCT and histomorphometry at day 7. Hematoxylin and eosin (H&E) staining and immunohistochemical staining were used to examine skin lesions on the calvaria at day 7. Ultra-pure LPS from *E. coli* (100 μg in 20 μL PBS/mouse) was injected every other day for 2 times. Mice were examined at day 4 by qPCR and at day 5 by microCT and histomorphometry. The intersection of the coronal and sagittal sutures was used as a reference position of injections.

### In vitro *Pg* cultures

*Pg* (ATCC 33277) was cultured on sheep blood agar plates and further grown in trypticase soy broth supplemented with yeast extract (5 g/L), hemin (5 mg/L), and menadione (50 mg/L) at 37 °C in an anaerobic condition (AnaeroPack, Mitsubishi). The density of *Pg* was determined by the absorbance at 600 nm. After centrifugation, supernatants were collected and filtered through a 0.2 μm filter. *Pg* was washed with PBS then heat-killed at 70 °C for 1 h followed by the suspension in the culture medium above. Absence of live *Pg* was confirmed by culturing the heat-killed *Pg*. *Pg* supernatants and heat-killed *Pg* were freshly prepared for every experiments.

### Calvaria injection of live *Pg*

*Pg* (2 × 10^9^ CFU in 20 μL growth media/mouse) was injected onto the calvaria of 10 to 11-week-old mice daily for 5 times (the first day of injection = day 1), and bone tissues were analyzed at day 6.

### Oral inoculation of live *Pg*

Ten to eleven-week-old mice were pretreated with Sulfatrim (200 mg sulfamethoxazole and 40 mg trimethoprim/5 mL) in drinking water for 7 days to reduce the amount of commensal oral bacterial flora. After a 3-day interval with Sulfatrim-free water (day 10), 2 × 10^9^ CFU of *Pg* (ATCC 33277) in 50 μL of 2% carboxymethyl cellulose (CMC)/PBS were inoculated onto the entire gingival tissue of the anesthetized mice every other day for 5 times using a malleable 24 G feeding needle. Control mice were inoculated with vehicle alone (2% CMC). Mice were examined by immunofluorescent staining for a *Pg* component at 3 days, qPCR and histomorphometry at 7 days, and microCT at 42 days after the last inoculation.

### MicroCT analysis

Calvarial bone tissue (day 7 for Pam3CSK4, *S. aureus* LTA, and *Pg* LPS. Day 6 for live *Pg*. day 5 for *E. coli* LPS) were fixed with 4% paraformaldehyde (PFA) for 24 h and soaked in 70% ethanol for scanning with the Skyscan1176 (Bruker) under the following conditions: 50 kV X-ray energy, 8.43 μm pixel size, and 0.3° rotation step with 926 ms of exposure time. Scanned data were reconstructed with NRecon software (Bruker) with the 0 to 0.18 dynamic range. Three-dimensional (3D) images were created by CTVox software (Bruker) based on the volume rendering method. Bone erosion area including sutures in 6 × 6 mm calvarial area centered at the intersection of the coronal and sagittal sutures (approximately 713 slices) was measured as pixels with ImageJ (NIH) and divided by the total number of pixels. The 6 × 6 mm area was used for quantitating bone surface/bone volume (BS/BV) and bone volume/tissue volume (BV/TV) by CTAnalyser (Bruker) with a threshold value of 48. Maxillae (at 42 days after the last inoculation) fixed with 4% PFA/PBS for 24 h were scanned with the Skyscan1176 with the 0 to 0.22 dynamic range. 3D images were created and the occlusal plane was aligned to parallel with the transverse plane using the DataViewer (Bruker). The total distance between cementoenamel junction (CEJ) and alveolar bone crest (ABC) at the underneath 12 cusps of the right maxillary three molars was measured in the reconstructed 2D images with CTAnalyser. The alveolar bone between two buccal roots underneath the second molar of the right maxilla that is composed of 10 slices was used for quantitation of BV/TV with a threshold value of 65. For in vivo microCT scanning of jawbones, mice were anesthetized with isoflurane and CEJ-ABC distances were measured by CTAnalyser with a 0.6° rotation step and a threshold value of 40.

### H&E staining and histomorphometric analysis

Calvariae and maxillae were fixed with 4% PFA for 24 h, decalcified with EDTA (0.5 M, pH 7.2), and embedded in paraffin. Six μm sections cut in the coronal plane were subjected to H&E and tartrate-resistant acid phosphatase (TRAP) staining. The number of osteoclasts/bone surface (N.Oc/BS) and osteoclast surface/bone surface (Oc.S/BS) on the calvarial bone surface 3 mm anterior and posterior to the intersection of the coronal and sagittal sutures or on the alveolar bone surface between the mesial and buccal roots underneath the right maxillary first molar were measured by Bioquant Osteo software in a blinded manner.

## Immunohistochemical and immunofluorescent staining

Paraffin sections were treated with 10 µg/mL of proteinase K (Gold Biotechnology) for 10 min at 37 °C for antigen retrieval and endogenous peroxidases were quenched with 3% $H_2O_2$/PBS solution. After blocking with 2% serum from the animal species in which secondary antibodies were raised, sections were incubated with antibodies against MYD88 (LS-C357983, LSBio), TLR2 (NB100-56720, Novus), TLR4 (ab13867, Abcam), F4/80 (Clone A3-1, Bio-Rad), Ly-6G (sc-53515, Santa Cruz Biotechnology) or *Pg* (DMAB9447, Creative Diagnostics) overnight at 4 °C. The M.O.M .kit (Vector Lab) was used for anti-*Pg* antibody. After washing with PBS, sections were incubated with biotinylated secondary antibodies (Vector Laboratories) for 60 min at room temperature and treated with VECTASTAIN® Elite ABC-HRP Kit (Vector Lab). After color development by ImmPACT DAB (Vector Lab) and sections were counter stained with methyl green or hematoxylin. For *Pg* component detection, Alexa Fluor 488 conjugated secondary antibody (Invitrogen) was used and counterstained with DAPI (Thermo Fisher).

## Isolation of osteoblast- and osteocyte-enriched cell populations

Calvariae were aseptically dissected from 10 to 11-week-old C57BL/6 J male mice and cut into 3 × 3 mm pieces. Pooled bone pieces were serially digested as described by ref. 31. using type I collagenase (300 U/mL, Worthington Biochemical) in α-MEM (Thermo Fisher) and EDTA (5 mM, pH = 7.4; ACROS Organics) in Ca- and Mg-free Hank's balanced salt solution (HBSS, Thermo Fisher) supplemented with 1% BSA (Research Product International). All digestion steps were performed in 3.5 mL solutions in 6-well plates rotated at 170 RPM under the condition of 5% $CO_2$ at 37 °C. After each digestion, digestion solution was collected, bone pieces were rinsed with HBSS for two times, then digestion solution and HBSS were combined together as the single fraction (F). The combined cell suspensions were spun down at 500 g for 5 min, resuspended in α-MEM supplemented with 10 % FBS and 1% penicillin/streptomycin (P/S) (Thermo Fisher), then each fraction was cultured for 48 h. For mechanistic studies, hematopoietic cells were removed from the pooled F1−4 cells and F6−9 cells using CD45 + cell depletion kit (Invitrogen). F5 was excluded from the experiments to better separate osteocytes from osteoblasts. The purified CD45-negative cells from F1–4 and F6−9 were regarded as the osteoblast-enriched cell population (Ob) and osteocyte-enriched cell population (Ocy), respectively. Ob and Ocy were suspended in α-MEM above, seeded on the 100 mm dishes. After 48 h, non-attached dead cells were removed, then Ob and Ocy were re-seeded on 8-well chamber slides (1 × 10^4 cells per well), 6-well plates (1.0 × 10^5 cells per well) or 60 mm dishes (5.0 × 10^5 cells) for further studies, including validation of the Ocy and Ob, comparison of the *Dmp1-Cre* specificity between Ocy and Ob (Supplementary Fig. 11d–g), and investigation of RANKL regulatory mechanisms in Ocy. Ocy was cultured on dishes coated with rat tail type I collagen (Cell Applications).

## Sclerostin and keratocan expression analysis in Ocy and Ob by immunofluorescent staining

Ocy and Ob were seeded on 8-well chamber slides (Corning) coated with or without type I collagen at a density of 1 × 10^4 cells per well. Cells were fixed with 4% PFA in PBS for 10 min at room temperature (RT) and gently washed with PBS for 5 min three times. After permeabilization with 0.2% Triton X-100 in PBS at RT for 10 min and washing with PBS for 5 min three times, cells were incubated with 2% BSA/PBS and 2% donkey or goat serum (Sigma-Aldrich) for 30 min at RT. Anti-sclerostin antibody (R&D systems, AF1589) or Anti-keratocan antibody (Abcam, ab128304) was applied for overnight at 4 °C with gentle shaking. After washing with PBS, cells were incubated with Alexa Fluor® 594 donkey

anti-goat antibody or Alexa Fluor® 594 goat anti-rabbit antibody (Thermo Fischer) for 1 h at RT. Normal goat or rabbit IgG was used for negative control. F-actin was visualized with Alexa Fluor 488 phalloidin (Invitrogen, A12379). Nuclei were stained with 4′,6-diamidino-2-phenylindole (DAPI). Fluorescent images were acquired with BZ-X800 microscope (Keyence, Osaka, Japan). The percentages of sclerostin- or keratocan-positive cells were calculated (the number of positive cells divided by the total number of cells) and averaged across five random fields per each well.

## Stimulation of Ob and Ocy

Ob and Ocy were stimulated with Pam3CSK4 (100 ng/mL), Pam2CSK4 (100 ng/mL), ultrapure LPS from *E. coli* (100 ng/mL), FLA-ST (100 ng/mL), or single-strand RNA (100 ng/mL) (InvivoGen).

## RNA isolation

RNA was extracted by RiboZol RNA extraction reagent (VWR). Calvarial bone tissue (6 × 6 mm) and skin tissue on calvaria (6 × 6 mm), both of which are centered at the intersection of the coronal and sagittal sutures, were used for RNA extraction. Maxillary gingival tissue at the palatal side (1 x 3 mm) and maxillary jawbone tissue including molars were used for RNA extraction. Calvarial and jawbone tissues were snap-frozen in liquid nitrogen after removing soft tissues, then crushed into powder using a tissue pulverizer (Cellcrusher Limited). Gingival tissues were homogenized by a tissue grinder (Thermo Fisher). Ob and Ocy were homogenized by pipetting.

## Reverse transcription-quantitative PCR (qPCR) analysis

Total RNA was isolated with Ribozol (VWR) and 1 µg of RNA was reverse transcribed using the High-Capacity cDNA reverse transcription kit (Life Technologies). qPCR was performed using PowerUP SYBR Green master mix and analyzed with the QuantStudio design & analysis software (Thermo Fisher). qPCR primers used in this study are listed in the Supplementary Table 1. Relative gene expression levels were calculated using a relative-standard curve method. Each gene expression levels were normalized by the expression levels of *Gapdh* except in Figure 9I and 9 M, where by *β-actin* was used for normalization.

## In vitro osteoclastogenesis

Bone marrow cells were harvested from the femur and tibia of 10 to 12-week-old *Myd88*^−/− mice. After the lysis of red blood cells with RBC lysis buffer (eBioscience), bone marrow cells were incubated in α-MEM supplemented with 10% FBS and P/S for 3 h on petri dishes to allow stromal cells to adhere to the dishes. Non-adherent cells were collected and seeded on 10-cm dishes and further incubated in the presence of M-CSF (25 ng/mL, PeproTech) for 2 days to obtain bone marrow-derived M-CSF-dependent macrophages (BMMs) as osteoclast precursor cells. Ob and Ocy in α-MEM supplemented with 10% FBS and P/S were cultured on 48-well plates at a density of 2.0 × 10^3 cells/well for 24 h, followed by the seeding of BMMs at a density of 2.0 × 10^4 cells/ well. After 8 h, co-cultured cells were stimulated with 100 ng/mL Pam3CSK4 or *E. coli* LPS in the presence of 25 ng/mL M-CSF (day 1). Culture medium containing M-CSF was replaced every 48 h and Pam3CSK4 or *E. coli* LPS were added to the cultures every 24 h. TRAP-positive (+) cells were visualized with the TRAP staining kit (Sigma-Aldrich) and TRAP + cells with more than 3 nuclei were counted as osteoclasts (TRAP + MNCs) at day 7.

## Inhibitor treatment of Ocy

U0126, SP600125, SB203580 (Cell Signaling Technology, CST), BMS-345541 (Sigma-Aldrich), 666-15, C188-9, STAT5-IN-1, T-5224, CADD522, H-89 (Med Chem Express), and T6167923 (Aobious) were used to block specific molecules or kinases. Cycloheximide (APExBIO) was used to

block protein synthesis. All inhibitors were added 2 h before the stimulation with TLR agonists.

## Western blotting

Cells were washed with ice-cold PBS and lysed with NP-40 lysis buffer (150 mM NaCl, 1% NP-40, 25 mM Tris-HCl (pH7.4), 5 mM EDTA, 10% Glycerol, 2.5 mM sodium pyrophosphate, 1 mM β-glycerophosphate) containing protease inhibitor cocktail (P8340, Sigma-Aldrich) and phosphatase inhibitors (P0044 and P5726, Sigma-Aldrich) on ice. Protein concentrations were measured by BCA Protein Assay Kit (Thermo Fisher). Proteins were boiled in Laemmli buffer for 5 min. Ten to Fifteen micrograms of protein per lane was separated on 7.5 or 10% of polyacrylamide gels, transferred to nitrocellulose membrane for Western blotting analysis. After blocking with 5 % non-fat skim milk in Tris-buffered saline solution containing Tween-20 (TBST), membranes were incubated with Can Get Signal solution (TOYOBO) containing a primary antibody overnight at 4 °C, followed by incubation with a corresponding HRP-conjugated secondary antibody (CST). Bands were detected using SuperSignal West Dura or Femto chemiluminescent substrate (Thermo Fisher) and visualized by the Celvin S 320+ Chemiluminescence Imager (Biostep). For quantitative comparisons, the samples were derived from the same experiment and blots were processed in parallel. Primary antibodies used in this study are listed in the Supplementary Table 2.

## Transduction of IDG-SW3 cells with lentivirus

IDG-SW3 cells (provided by Lynda Bonewald, Indiana University) were cultured at 33 °C in the presence of IFN-γ (50 U/mL, Gibco). Lentivirus containing mouse *Fbxl19*, mouse *Pdlim2*, mouse *Fbxl19* shRNA, or mouse *Pdlim2* shRNA and negative control lentiviruses were purchased from GeneCopoeia. $1.0 \times 10^5$ IDG-SW3 cells were seeded on the 60 mm dishes coated with rat tail type I collagen and infected with lentiviruses at a multiplicity of infection (MOI) of 5 in the presence of polybrene (8 μg/mL). After 72 h, puromycin selection (5 μg/mL) was started. IDG-SW3 cells resistant to puromycin were expanded and cultured at 37 °C for 28 days in the absence of IFN-γ to differentiate into mature osteocyte-like cells, then stimulated with Pam3CSK4.

## siRNA transfection into MLO-Y4 cells

MLO-Y4 cells (provided by Lynda Bonewald, Indiana University) were seeded on 6-well plates coated with rat tail type I collagen ($5 \times 10^4$ cells per well) and cultured in α-MEM supplemented with 2.5% FBS, 2.5% calf serum, and 1% P/S at 37 °C under 5% CO2. TransIT-X2® Dynamic Delivery System (Mirus) were used for the transfection of siRNAs (25 nM each). After 24 h, medium was replaced with antibiotics-free medium. Complexes of siRNAs or control siRNAs in TransIT-X2 were added to the medium. After 48 h of siRNA transfection, cells were harvested for qPCR analyses.

## Immuno-coprecipitation

Cells were washed twice with ice-cold PBS and lysed with NP-40 lysis buffer. After pre-clearing with protein A/G-Plus-Agarose (Santa Cruz Biotechnology), 300 μg of total protein was incubated with 1 μg of anti-CBP antibody (#7389, CST), anti-CREB antibody (#9197, CST), and anti-STAT3 antibody (#12640, CST) at 4 °C overnight, then incubated with 20 μL of protein A/G-Plus-Agarose for additional 3 h. After washing three times with lysis buffer, precipitated agarose was resuspended in Laemmli buffer and boiled for 5 min. Samples were subjected to SDS-PAGE followed by Western blotting.

## Cleavage under targets and release using nuclease (CUT & RUN) assay

All procedures were performed according to the manufacturer's protocol (CUT & RUN assay kit, CST). Briefly, $2 \times 10^5$ cells from Ocy were

resuspended in wash buffer (20 mM HEPES-NaOH pH 7.5, 150 mM NaCl, 0.5 mM spermidine, and protease inhibitor cocktail), concanavalin A-magnetic beads added, then rotated for 10 min at room temperature. Cell-bead conjugates were resuspended in 200 μL of digitonin buffer (wash buffer with 2.5% digitonin solution) containing 2 μg of anti-CBP (#7389, CST), anti-CREB (#9197, CST), anti-STAT3 (#12640, CST), anti-CRTC2 (MAB6338, R&D) primary antibody, or rabbit IgG (# 66362, CST), rotated overnight at 4 °C, resuspended in 250 μL of antibody buffer and 7.5 μL of pAG-MNase enzyme (# 57813, CST), followed by the rotation at 4 °C for 1 h. After washing with digitonin buffer, ice-cold 150 μL of digitonin buffer containing $CaCl_2$ was added, and incubated on ice for 30 min followed by the addition of 150 μL of stop buffer containing 5 ng *S. cerevisiae* spike-in DNA used for sample normalization. After incubation at 37 °C for 15 min, samples were centrifuged at 16,000 g for 2 min at 4 °C. Tubes were placed on the magnetic rack, then supernatants were collected. DNA was purified using the DNA purification buffers and spin columns (#14209, CST). qPCR was conducted using primer sets in the Supplementary Table 3 to quantitate the amount of DNA at the *Rankl* promoter and enhancer regions. Data were calculated using the % input method for each antibody and represented as the fold enrichment relative to the isotype control (rabbit IgG).

## ELISA

Soluble-form RANKL levels in cell culture supernatants were measured by using a murine sRANKL pre-coated ELISA kit (Biogems). Fluorescence intensities were acquired with SpectroMax i3 and analyzed with SoftMax Pro6 (Molecular Devices).

## In vivo MYD88 inhibitor administration

Mice were intraperitoneally administered with T6167923 (40 mg/kg) every other day starting from the day of last *Pg* inoculation until the day of analysis.

## Statistics

The two-tailed Student's *t*-test was used to compare means between two groups. One-way ANOVA with the Tukey–Kramer post-hoc test was used to compare means among three or more groups. $p < 0.05$ was considered significant. GraphPad Prism software was used for statistical analysis. Male and female groups were analyzed independently to detect sexual dimorphisms.

## Reporting summary

Further information on research design is available in the Nature Research Reporting Summary linked to this article.

## Data availability

All data supporting the findings of this study are available within this article and its supplementary information file. Source data are provided with this paper.

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

## Acknowledgements

T.Y., M.K., and Y.U. thank Tianli Zhu at the histology core of Indiana University School of Dentistry for assisting with sample preparation. This work was supported by the National Institute of Aging (P01AG039355) to L.F.B. and National Institute of Dental and Craniofacial Research (R01DE025870, R01DE025870-06S1, R21DE030561) to Y.U.

## Author contributions

T.Y. and Y.U. conceived the overall hypothesis and designed the experimental plans. T.Y., M.K., A.A.P.D., R.U., and Y.U. performed the experiments. M.P. and L.F.B. contributed to establishing a method for separately isolating primary osteoblast-enriched and osteocyte-

enriched cell populations. E.M.G. contributed to evaluating the data quality and reproducibility. R.E.R., C.V.H., and W.H.B. created and provided the TLR2 conditional knockout mice. T.Y., M.K., M.P., L.F.B., E.M.G., and Y.U. interpreted the experimental data. T.Y. and Y.U. wrote the manuscript and all authors reviewed and discussed the manuscript. Y.U. supervised the study.

## Competing interests

The authors declare no competing interests.
