## [Peer Review File · Nature Communications]

Osteocytes directly regulate osteolysis via MYD88 signaling in bacterial bone infectionREVIEWER COMMENTS

Reviewer #1 (Remarks to the Author):

In this manuscript, Yoshimoto et al study TLR signaling in osteocytes in the setting of periodontal infections. Previous studies have suggested that osteocytes are an important cellular source of the osteoclastogenic factor RANKL in normal bone remodeling and in periodontitis. These observations are extended here, along with new mechanistic data which focusing on downstream signaling pathways linking TLR activation to RANKL gene induction via MYD88. Interestingly, TLR-induced RANKL gene regulation in osteocytes involves distinct signaling pathways than hormonal RANKL induction. Therefore, this study does not establish new paradigms, but rather represents a largely solid mechanistic advance to this field. Currently, enthusiasm is limited due to a heavy reliance on the DMP1-Cre transgene to make claims about 'osteocyte-specific' biology, and the cursory nature of the signaling studies examining PDLIM2 in STAT3 regulation downstream of TLR/MYD88 signaling.

Major points:

1. Since it is already well-established that osteocytes can produce RANKL downstream of TLR activation, the major novel aspect of this finding is the addition of MYD88-dependent signaling pathways. This isn't entirely surprising since MYD88 is a key adaptor protein downstream of TLR signaling in multiple cell types in the immune system. For this reason, particular emphasis on cell type-specific events downstream of MYD88 is important. The authors present data that MYD88 signaling (somehow) targets ubiquitin ligases FBXL19 and PDLIM2 to regulate CREB and STAT3 levels, respectively. However, these data (Figure 7) rely heavily on overexpression studies in cultured cells. The authors should employ loss of function (shRNA or CRISPR knockout) approaches in cultured osteocytes or (ideally) delete these ubiquitin ligases in vivo and then assess TLR-induced RANKL upregulation and periodontal bone loss. Without data along these lines it is very difficult to evaluate the physiologic significance of this model.
2. As acknowledged in the discussion, DMP1-Cre is an imperfect tool to achieve 'osteocyte-specific' deletion. Recent data has demonstrated DMP1-Cre activity in multiple non-skeletal cells (PMID 27237054 and 34725346, for example). This is most important for Figure 3 where the MYD88 LSL model is used. Here, it is quite important to demonstrate that DMP1-Cre does not restore MYD88 expression in innate and adaptive immune cells. Without this data, the conclusion that osteocytes and osteoblasts are the major cells that respond to Pg and Pam3 and then produce RANKL is simply an assumption.
3. The studies performed in Figure 5 rely heavily on pharmacologic inhibitors. While these are helpful tools, results should be confirmed/extended using genetic approaches. For example, signaling studies could be performed using osteocytes from Dmp1-Cre ; MYD88 conditional knockout mice (or osteocytes obtained from MYD88 floxed mice treated with Cre ex vivo). In addition, it would be ideal to use shRNA or CRISPR approaches to delete the kinases/TFs studied in Figure 5b-c.

Minor points:

1. Page 6, line 7- wording of this sentence is awkward, could change to "T and B lymphocytes are well-known cellular sources of RANKL in periodontitis."
2. Figure 2a- the text should explain that RAG1^{-/-} mice lack lymphocytes.
3. The relevance of the calvarial resorption model to periodontitis studies should be explained.
4. Figure S6g shows very interesting results. The authors should consider to move this to the main figures.
5. For Figure 6c, it is unclear how the 3 control samples can have exactly the same value. Raw data is needed to review methodology for quantifying immunoblots.
6. A diagram showing the RANKL gene locus and position of different primer positions would be help for Figure 5g.
7. For Figure 7g, a western blot is needed to demonstrate how much FBXL19 for overexpressed (versus endogenous levels) in this experiment. Overall, the data in Figure 7 showing effects of PDLIM2 on STAT3 is not as strong as the results suggesting that FBXL19 controls CREB levels. In particular, the blot in Fig 7j is difficult to interpret. Results for PDLIM2 similar to Fig 7e for FBXL19

are needed.

8. For Figure 8, more details are needed for the in vivo study. Was DMSO really used as a solvent for the MYD88 inhibitor for in vivo treatments? Generally, aqueous solvents are needed for 'translational' studies. How do the authors interpret the data in Figure 8b which shows loss of alveolar bone in Pg-infected mice treated with the MYD88 inhibitor?

Marc Wein

Reviewer #2 (Remarks to the Author):

The manuscript by Yashimoto et al. provides for a deeper mechanistic understanding of the role played by osteocytes in the context of microbial-influenced bone activity. Although the manuscript focuses on infection related to the oral cavity, it is felt strongly that the findings will be of multi-disciplinary value. The well organized and conducted studies are rationalized in the context of clinical observation, and build on and incomplete understanding that MyD88 as a signaling conduit for innate immune sensing of microbial patterns in bone turnover as it relates to regulation of osteolysis. The title speaks to bone infection; however bone infection (bacteria in bone) is not directly examined in this study using live bacterial challenge – overall the lack of confirming the outlined mechanisms suffers from a lack of use of live bacteria beyond the *P. gingivalis* oral infection model – which it appears may not be linked to direct infection of bone, but rather a complex inflammatory signature that is reported to be dependent on TLR2 – at least in animal models.

The area of osteolysis has been unable to advance significantly due in part to a clear picture of osteocyte-mediated effects on bone health. The identification of MyD88-mediated system controlling osteolysis is important – however, the limited consideration of live bacteria and almost complete reliance on PAMPs does limit enthusiasm. It is appreciated that PAMPs serve as a tool in a reduction approach to detail mechanism of action, and as very elegantly laid out in this manuscript, the use of several cutting-edge (targeted gene restoration, etc.) and gold-standard approaches combined with rigorous and multi-modal considerations for mechanism characterization are a strength of this manuscript. However, the use of PAMPs alone does not in most cases recapitulate the host sensing of a microbe during infection. The focus on RANKL is appropriate and the interpretation of CREB and STAT3 influencing ubiquitination via FBXL19 and PDLIM2 is supported by presented data. Of high importance and a major finding is that osteocytes and not T/B cells are the primary source of RANKL. Of note is the anticipated result that use of the MyD88 inhibitor T6167923 partially influences the oral bone loss in the *P. gingivalis* infection model is expected as prior studies have shown that TLR2 is a key molecule in this process and only signals via MyD88, thus the importance of these findings are felt more confirmatory than reinforcing. However, it is important that this finding occur in vivo. More examination of the oral architecture in the context of the oral infection seems warranted to confirm the very elegant in vitro studies. Overall, there is enthusiasm for this manuscript; however, the sparse use of live bacteria represents a limitation to the findings.

Reviewer #3 (Remarks to the Author):

The goal of this study was to examine the role of Myd88 in the bone loss caused by bacterial infection. The authors find that deletion of Myd88 using a Dmp1-Cre driver strain blocked bone loss and stimulation of RANKL production in an infection model. Post-translational modification of Creb and Stat3 were found to mediate the stimulation of RANKL levels and an Myd88 inhibitor blunted bone loss in an infection model. Based on these results, the authors conclude that Myd88 in osteocytes directly regulates bone resorption during infection. Overall, the amount of work is impressive and many of the results are convincing.

Comments.

1. Although the authors acknowledge that the Dmp1-Cre driver strain is not specific for osteocytes,

this acknowledgement comes late in the discussion and is not reflected in the title or anywhere else in the manuscript. This is especially important as Dmp1-Cre has been shown to target CAR cells and CAR cells have been shown to be an important source of RANKL (JBMR 31:2001, 2016 and JCI 131(2):e140214, 2021). The authors should either provide additional functional evidence that the effects of Myd88 deletion are the result of loss in osteocytes or they should temper their conclusion that the effect is solely due to loss in osteocytes.

2. The CREB and STAT3 inhibitor studies presented in Figure 5C are not compelling. For example, in the first panel, the levels of RANKL change significantly in response to different levels of PBS. It is also unclear what the values of PBS refer to; it is unlikely that they refer to the micromolar concentration of PBS in the culture. Nonetheless, it is unclear why any level of PBS would suppress RANKL levels by 4-fold. Because CREB and STAT3 are central players in the model proposed by the authors, their role in RANKL regulation should be confirmed using mRNA knockdown studies or CRISPR-mediated gene inactivation.

3. Figure 1e shows that Pam3 suppresses OPG expression in bone. While the suppression of OPG may not be mediated by Myd88, it may still contribute to the increase in bone resorption. The authors should consider the possibility that changes in RANKL may not be the only mechanism responsible for the increase in resorption.

4. The study ends with a translational experiment aiming to show that an Myd88 inhibitor blocks inflammatory bone loss. However, results from this important experiment are too limited. At a minimum, a gene expression analysis of bone tissue showing changes in RANKL, OPG, and osteoclast marker genes would seem important. Preferably, histology of osteoclasts should be performed. This is important since the mice still lost alveolar bone even in the presence of the inhibitor.

Dear Reviewers,

August 14, 2022

We thank each reviewer for carefully reading our manuscript. Also, we very much appreciate your comments on our manuscript, all of which are very fruitful and completely agreeable to us. The revisions we have made in response to the comments from each Reviewer are as follows:

Reviewer #1

In this manuscript, Yoshimoto et al study TLR signaling in osteocytes in the setting of periodontal infections. Previous studies have suggested that osteocytes are an important cellular source of the osteoclastogenic factor RANKL in normal bone remodeling and in periodontitis. These observations are extended here, along with new mechanistic data which focusing on downstream signaling pathways linking TLR activation to RANKL gene induction via MYD88. Interestingly, TLR-induced RANKL gene regulation in osteocytes involves distinct signaling pathways than hormonal RANKL induction. Therefore, this study does not establish new paradigms, but rather represents a largely solid mechanistic advance to this field.

We agree that this study may not establish a new paradigm in the field of osteocyte biology. However, to the best of our knowledge, previous studies have *never* demonstrated the *in vivo* impact of the osteocyte MYD88 signaling pathway on osteolysis associated with infections, even though some *in vitro* studies using immortalized cell lines suggested it. Also, we appreciate that the reviewer recognized our discoveries of new mechanisms of RANKL induction downstream of MYD88, which is different from the previously-identified mechanisms induced by PTH.

Currently, enthusiasm is limited due to a heavy reliance on the Dmp1-Cre transgene to make claims about 'osteocyte-specific' biology, and the cursory nature of the signaling studies examining PDLIM2 in STAT3 regulation downstream of TLR/MYD88 signaling.

We completely agree that our genetic studies heavily rely on the *Dmp1-Cre* mice. To carefully check the *Dmp1-Cre* specificity, we improved a previous procedure and established a method to differentially isolate osteoblast- and osteocyte-enriched cell populations from adult mouse calvariae (Supplementary Fig. 11). Calvariae were used as a source of these cells because we employed bacterially-induced osteolysis models at the craniofacial and oral regions in the current study. Remarkably, comparing the deletion or restoration efficiency of MYD88, TLR2, or RANKL protein (by Western blotting) showed that *Dmp1-Cre* is highly specific to the osteocyte-enriched cell population (referred as Ocy in the manuscript) than to the osteoblast-enriched cell population (referred as Ob in the manuscript) in calvarial bone (Supplementary Fig. 11g).

We, however, definitely acknowledge that previous elegant studies using Ai9 or Ai14 reporter mice (Zhang and Link 2016 JBMR, Lim 2017 Bone Res, Wang et al 2021 Nat Commun) showed that *Dmp1-Cre* is also active in mature osteoblasts of long bones. We also recognize that our method of differentially isolating osteocytes from osteoblasts is still suboptimal and imperfect. Therefore, our data were interpreted to indicate that the TLR-MYD88 pathway in both osteocytes and mature osteoblasts regulates bacterially-induced osteolysis in our oro-craniofacial osteolysis models due to bacterial PAMPs. However, considering **1)** our new discovery that Ocy induces significantly more RANKL than Ob in response to PAMPs (Fig. 4, Supplementary Fig. 12), **2)** the data that *Dmp1-Cre* is highly specific to the Ocy of calvariae (Supplementary Fig. 11g), and **3)** the fact that osteocytes are the most abundant cell type in the bone, our results suggest that osteocytes are playing a dominant role in the mechanism. These points were discussed in the discussion section. Furthermore, in addition to *Dmp1-Cre*, we have analyzed the *Osteocalcin-Cre;Myd88^{fl/fl}* mice and discovered that the *Osteocalcin-Cre;Myd88^{fl/fl}* mice, which deletes MYD88 in both Ocy and Ob, protect against osteolysis equivalent to that in *Dmp1-Cre* mice. The result is also discussed (Supplementary Fig. 20).

The role of PDLIM2 in STAT3 regulation downstream of TLR/MYD88 signaling has been validated using a PDLIM2 knockdown approach in IDG-SW3 cells (Supplementary Figure 17).

Major points:

1. Since it is already well-established that osteocytes can produce RANKL downstream of TLR activation,

the major novel aspect of this finding is the addition of MYD88-dependent signaling pathways. This isn't entirely surprising since MYD88 is a key adaptor protein downstream of TLR signaling in multiple cell types in the immune system. For this reason, particular emphasis on cell type-specific events downstream of MYD88 is important.

We completely agree on the comment. This is the reason why investigated the more detailed mechanisms of RANKL induction downstream of the MYD88 pathway. However, no previous studies have demonstrated the *in vivo* impact of the MYD88 pathway in osteocytes/mature osteoblasts on the skeleton, although *in vitro* studies implicated it. I hope the reviewer also recognizes the significance of our genetic studies that proved and established the importance of the osteocyte MYD88-RANKL axis *in vivo*. Further, we employed the primary osteocytic cells (Ocy) vigorously in order to determine the molecular events in osteocytes downstream of MYD88, because almost all previous *in vitro* discoveries were obtained by using immortalized cell lines that might not completely recapitulate the osteocyte phenotypes.

The authors present data that MYD88 signaling (somehow) targets ubiquitin ligases FBXL19 and PDLIM2 to regulate CREB and STAT3 levels, respectively. However, these data (Figure 7) rely heavily on overexpression studies in cultured cells. The authors should employ loss of function (shRNA or CRISPR knockout) approaches in cultured osteocytes or (ideally) delete these ubiquitin ligases *in vivo* and then assess TLR-induced RANKL upregulation and periodontal bone loss. Without data along these lines it is very difficult to evaluate the physiologic significance of this model.

Accordingly, using shRNA lentiviruses, we knocked down *Fbxl19* and *Pdlim2* in the IDG-SW3 cell line. We found that knocking down these ubiquitin ligases decreases CREB/CBP and STAT3 ubiquitination, increases CREB/CBP and STAT3 protein, and promotes *Rankl* expression and induction, which are opposite to the outcomes from overexpression studies (Supplementary Fig. 16 and 17). To be honest, we also recognize that deleting these ubiquitin ligases *in vivo* is a critical approach, but we have not created the mice yet. We are planning to obtain or generate these floxed mice for further *in vivo* analysis. I expect that results from knockdown approaches in IDG-SW3 cell will satisfy the reviewer's request.

2. As acknowledged in the discussion, DMP1-Cre is an imperfect tool to achieve 'osteocyte-specific' deletion. Recent data has demonstrated DMP1-Cre activity in multiple non-skeletal cells (PMID 27237054 and 34725346, for example). This is most important for Figure 3 where the MYD88 LSL model is used. Here, it is quite important to demonstrate that DMP1-Cre does not restore MYD88 expression in innate and adaptive immune cells. Without this data, the conclusion that osteocytes and osteoblasts are the major cells that respond to Pg and Pam3 and then produce RANKL is simply an assumption.

We appreciate the comment from the reviewer. To check the off-target recombination of *Dmp1-Cre*, we performed genomic PCR using DNAs isolated from bone and innate and adaptive immune cells/tissues. No obvious off-target deletion or restoration of the *Myd88* gene by *Dmp1-Cre* has been detected in these immune cells/tissues (Supplementary Fig. 2, 10), showing that osteocytes and mature osteoblasts are the primary tissue that initially and directly respond to PAMPs to produce RANKL and trigger inflammatory reactions in *Dmp1-Cre;Myd88^{sl/sl}* mice.

3. The studies performed in Figure 5 rely heavily on pharmacologic inhibitors. While these are helpful tools, results should be confirmed/extended using genetic approaches. For example, signaling studies could be performed using osteocytes from *Dmp1-Cre* ; MYD88 conditional knockout mice (or osteocytes obtained from MYD88 floxed mice treated with Cre *ex vivo*).

According to the Reviewer's suggestion, we isolated the Ocy from *Dmp1-Cre;MYD88^{fl/fl}* mice and performed signaling studies to compare the CREB and STAT3 activation. However, because lack of MYD88 decreases CREB and STAT3 proteins (the mechanisms are presented in the next section of the manuscript), total amounts of CREB and STAT3 protein are not consistent between Cre-positive Ocy and Cre-negative Ocy. Therefore, we concluded that showing this result at this point of the manuscript might confuse the readers. Accordingly, we kept the original data using the MYD88 inhibitor in Fig. 5e. The result from Ocy was presented in the Supplementary Fig. 13e.

In addition, it would be ideal to use shRNA or CRISPR approaches to delete the kinases/TFs studied in Figure 5b-c.

The reason why we used pharmacological inhibitors is that we have experienced difficulties in transfecting siRNAs/shRNA plasmids or infecting shRNA viruses in Ocy efficiently. Alternatively, we used the MLO-Y4 osteocytic cells that also induce RANKL in response to PAMPs. We have transfected siRNAs into MLO-Y4 cells and found that knocking down ERK, CREB, or STAT3 reduces *Rankl* induction (Supplementary Fig. 13a-d). I expect that the reviewer understands the difficulties in manipulating gene expression in primary osteocytic cells.

Minor points:

1. Page 6, line 7- wording of this sentence is awkward, could change to "T and B lymphocytes are well-known cellular sources of RANKL in periodontitis."

Thank you very much for the comment. We have changed the wording as suggested.

2. Figure 2a- the text should explain that RAG1^{-/-} mice lack lymphocytes.

"that lack T/B lymphocytes" has been added after *Rag1^{-/-}* mice.

3. The relevance of the calvarial resorption model to periodontitis studies should be explained.

We intended to use the calvarial model as one of the bone infection models different from periodontitis. However, thanks to the Reviewer 2's suggestion, we have now injected live *P.gingivalis* (*Pg*) and *Pg* LPS onto the calvaria and confirmed that *Dmp1-Cre/Myd88^{fl/fl}* mice protect against osteolysis caused by these natural *Pg*-derived PAMPs (Supplementary Fig. 4, 5). The results with *Pg*-derived PAMPs in calvarial model will serve as the rationale of studying another *Pg*-induced osteolysis model, periodontitis.

4. Figure S6g shows very interesting results. The authors should consider to move this to the main figures.

Fig. S6g has been moved and presented as the main Fig. 2g.

5. For Figure 6c, it is unclear how the 3 control samples can have exactly the same value. Raw data is needed to review methodology for quantifying immunoblots.

We apologize to the reviewer for our oversight. We performed three independent quantitation, and the average of intensities of bands from *Dmp1-Cre* (-);*Myd88^{fl/fl}* mice were set as 100%. Accordingly, 3 flat circles in the left column have been removed and SD has been added.

6. A diagram showing the RANKL gene locus and position of different primer positions would be help for Figure 5g.

Thank you very much for the suggestion. A diagram has been added in the Fig. 5g.

7. For Figure 7g, a western blot is needed to demonstrate how much FBXL19 for overexpressed (versus endogenous levels) in this experiment.

A Western blot of FBXL19 was presented in the original 7g (whole cell lysates). Nonetheless, we quantitated the FBXL19 protein levels and found that lentiviral overexpression increased the FBXL19 protein approximately 3-fold. Similarly, PDLIM2 protein levels were quantitated in the Fig. 7l.

Overall, the data in Figure 7 showing effects of PDLIM2 on STAT3 is not as strong as the results suggesting that FBXL19 controls CREB levels. In particular, the blot in Fig 7j is difficult to interpret.

Fig. 7k (previously Fig. 7j) shows that the PDLIM2-STAT3 interaction decreases in Ocy when stimulated with Pam3CSK4, which is similar to that of FBXL19-CREB/CBP. We have used the image with better contrast.

Results for PDLIM2 similar to Fig 7e for FBXL19 are needed.

A Western blot for PDLIM2 has been added as the new Fig. 7j.

8. For Figure 8, more details are needed for the in vivo study. Was DMSO really used as a solvent for the MYD88 inhibitor for in vivo treatments? Generally, aqueous solvents are needed for 'translational' studies.

Thank you very much for the suggestion. We have added the histomorphometry and qPCR data in Fig. 8. We understand the concern about the solvent of the MYD88 inhibitor. Because T6167923 cannot be dissolved in PBS, DMSO was needed as a solvent for *in vivo* administration as well as for *in vitro* studies. In this regard, our MYD88 inhibitor administration may not be purely considered as a translational treatment. However, the fact is that T6167923 in DMSO works well for protecting against jawbone resorption at least in mice. We will test other solvents which may be more suitable for translational administration (e.g., corn oil) in the future. We are also contemplating testing another MYD88 inhibitor ST2825, which is orally active but still DMSO is required to dissolve.

How do the authors interpret the data in Figure 8b which shows loss of alveolar bone in Pg-infected mice treated with the MYD88 inhibitor?

The comparison we want the reviewer to look at is between DMSO-treated *Pg*-infected mice vs. T6167923-treated *Pg*-infected mice (red vs. blue column), which showed the increase of BV/TV. We also noticed that, while T6167923 completely rescued an increase in the CEJ-ABC distance, it partially rescued a decrease in BV/TV. The difference in the magnitude of the effect of the MYD88 inhibitor suggests that the therapeutic response to the MYD88 inhibitor depends on the parameter and site of the alveolar bone.

Reviewer #2

The manuscript by Yoshimoto et al. provides for a deeper mechanistic understanding of the role played by osteocytes in the context of microbial-influenced bone activity. Although the manuscript focuses on infection related to the oral cavity, it is felt strongly that the findings will be of multi-disciplinary value. The well organized and conducted studies are rationalized in the context of clinical observation, and build on and incomplete understanding that MyD88 as a signaling conduit for innate immune sensing of microbial patterns in bone turnover as it relates to regulation of osteolysis. *The title speaks to bone infection*; however bone infection (bacteria in bone) is not directly examined in this study using live bacterial challenge – overall the lack of confirming the outlined mechanisms suffers from a lack of use of live bacteria beyond the *P. gingivais* oral infection model – which it appears may not be linked to direct infection of bone, but rather a complex inflammatory signature that is reported to be dependent on TLR2 – at least in animal models. The area of osteolysis has been unable to advance significantly due in part to a clear picture of osteocyte-mediated effects on bone health. The identification of MyD88-mediated system controlling osteolysis is important – however, the limited consideration of live bacteria and almost complete reliance on PAMPs does limit enthusiasm. It is appreciated that PAMPs serve as a tool in a reduction approach to detail mechanism of action, and as very elegantly laid out in this manuscript, the use of several cutting-edge (targeted gene restoration, etc.) and gold-standard approaches combined with rigorous and multi-modal considerations for mechanism characterization are a strength of this manuscript. However, the use of PAMPs alone does not in most cases recapitulate the host sensing of a microbe during infection.

We completely agree with the reviewer's point. In addition to oral infection of *live P. gingivais* (Fig. 2), calvarial bone was infected with *live P. gingivais* following the procedure by Zubery Y et al. (Infection and immunity 1998, 4158-4162). We discovered that the osteocyte/mature osteoblast MYD88 pathway also regulates osteolysis caused by live *P. gingivais*. Furthermore, we employed *Staphylococcus aureus*-derived lipoteichoic acid (LTA) and *P. gingivais*-derived LPS in the calvarial model, and found that *Dmp1-Cre;Myd88^{fl/fl}* mice are also protected from the osteolysis. Since these LTA and LPS are *natural* PAMPs from live bacteria, not chemically synthesized unlike Pam3CSK4, these findings support that activation of the osteocyte/mature osteoblast MYD88 pathway is a key mechanism for osteolysis when bone and surrounding tissues are infected by bacteria. These results are presented in the Supplementary Fig. 4 and 5.

The focus on RANKL is appropriate and the interpretation of CREB and STAT3 influencing ubiquitination via FBXL19 and PDLIM2 is supported by presented data. Of high importance and a major finding is that osteocytes and not T/B cells are the primary source of RANKL. Of note is the anticipated result that use of the MyD88 inhibitor T6167923 partially influences the oral bone loss in the *P. gingivais* infection model is expected as prior studies have shown that TLR2 is a key molecule in this process and only signals via

MyD88, thus the importance of these findings are felt more confirmatory than reinforcing. However, it is important that this finding occur in vivo. More examination of the oral architecture in the context of the oral infection seems warranted to confirm the very elegant in vitro studies.

We appreciate that the reviewer recognized the importance of our *in vivo* genetic studies. According to the reviewer's suggestion, other bone-resorption parameters in *Pg* periodontitis mice treated with T6167923 were examined. The data showed that systemic T6167923 administration blocked alveolar bone loss by inhibiting osteoclast formation. Results were presented in Fig. 8. We also thank the careful examination of the data by the reviewer. Currently, we do not have clear explanations as to why T6167923 treatment showed partial rescue from BV/TV, but not the CEJ-ABC distance. T6167923 might reach to osteocytes close the bone surface more efficiently than those located deep inside of the bone.

Overall, there is enthusiasm for this manuscript; however, the sparse use of live bacteria represents a limitation to the findings.

We appreciate the reviewer's enthusiasm for our discoveries. As mentioned above, in addition to *E.coli* LPS injection and oral live *Pg* infection, live *Pg*, *Pg* LPS, *S.aureus* LTA, all of which are natural bacterial PAMPs, were injected onto calvaria to reinforce our conclusions that the MYD88 pathway in osteocytes/mature osteoblasts regulates osteolysis in bacterial bone infection (Supplementary Fig. 4, 5).

Reviewer #3

The goal of this study was to examine the role of Myd88 in the bone loss caused by bacterial infection. The authors find that deletion of Myd88 using a Dmp1-Cre driver strain blocked bone loss and stimulation of RANKL production in an infection model. Post-translational modification of Creb and Stat3 were found to mediate the stimulation of RANKL levels and an Myd88 inhibitor blunted bone loss in an infection model. Based on these results, the authors conclude that Myd88 in osteocytes directly regulates bone resorption during infection. Overall, the amount of work is impressive and many of the results are convincing.

Comments.

1. Although the authors acknowledge that the Dmp1-Cre driver strain is not specific for osteocytes, this acknowledgement comes late in the discussion and is not reflected in the title or anywhere else in the manuscript. This is especially important as Dmp1-Cre has been shown to target CAR cells and CAR cells have been shown to be an important source of RANKL (JBMR 31:2001, 2016 and JCI 131(2):e140214, 2021). The authors should either provide additional functional evidence that the effects of Myd88 deletion are the result of loss in osteocytes or they should temper their conclusion that the effect is solely due to loss in osteocytes.

We absolutely agree with the concern about the *Dmp1-Cre* specificity. Accordingly, conclusion has been tempered. We concluded that the MYD88 pathway in both osteocytes and mature osteoblasts plays a critical role in the mechanism of osteolysis in the current infection models.

However, considering **1)** our new discovery that Ocy induces significantly more RANKL than Ob in response to PAMPs (Fig. 4, Supplementary Fig. 12), **2)** the data that *Dmp1-Cre* is highly specific to the Ocy of calvariae when deleting/restoring MYD88, TLR2, or RANKL (Supplementary Fig. 11g), and **3)** the fact that osteocytes are the most abundant cell type in the bone, our results suggest that osteocytes are playing a dominant role in the mechanism. Therefore, our data were interpreted to indicate that the TLR-MYD88 pathway in both osteocytes and mature osteoblasts, but likely dominantly the pathway in osteocytes, regulates bacterially-induced osteolysis in the current osteolysis models. These points were discussed in the discussion section.

2. The CREB and STAT3 inhibitor studies presented in Figure 5C are not compelling. For example, in the first panel, the levels of RANKL change significantly in response to different levels of PBS. It is also unclear what the values of PBS refer to; it is unlikely that they refer to the micromolar concentration of PBS in the culture. Nonetheless, it is unclear why any level of PBS would suppress RANKL levels by 4-fold.

We apologize for the confusing Figure 5C. In Fig. 5C, the numbers underneath the X axis represent the concentration of each inhibitor, not PBS. Labeling has been updated to avoid this type of misreading. We also thank the reviewer for catching the 4-fold *Rankl* suppression by the CREB inhibitor when stimulated with PBS. We assume that a certain level of CREB activity is required to maintain substantial RANKL expression of in osteocytes.

Because CREB and STAT3 are central players in the model proposed by the authors, their role in RANKL regulation should be confirmed using mRNA knockdown studies or CRISPR-mediated gene inactivation.

We found it was difficult to efficiently transfected siRNA or plasmids into primary osteocytes or differentiated (mineralized) IDG-SW3 cells. Therefore, CREB and STAT3 have been knocked down by siRNA in the osteocytic cell line MLO-Y4 that also induces *Rankl* in response to bacterial PAMPs. Results confirmed that these transcription factors are involved in the mechanism of *Rankl* induction downstream of MYD88 (Supplementary Fig. 13).

3. Figure 1e shows that Pam3 suppresses OPG expression in bone. While the suppression of OPG may not be mediated by Myd88, it may still contribute to the increase in bone resorption. The authors should consider the possibility that changes in RANKL may not be the only mechanism responsible for the increase in resorption.

We truly appreciate that the reviewer noticed the *Opg* decreases in the calvarial model. Actually, we knew the reduction. However, we did not mention this observation in the original manuscript because the main focus of the current study is *Rankl* expression. We mentioned to the *Opg* phenotype in the result section and discussed the possible involvement of OPG reduction in the mechanism of MYD88-mediated osteolysis in the discussion section.

4. The study ends with a translational experiment aiming to show that an Myd88 inhibitor blocks inflammatory bone loss. However, results from this important experiment are too limited. At a minimum, a gene expression analysis of bone tissue showing changes in RANKL, OPG, and osteoclast marker genes would seem important. Preferably, histology of osteoclasts should be performed. This is important since the mice still lost alveolar bone even in the presence of the inhibitor.

qPCR for osteoclast-associated genes and histomorphometry of osteoclast parameters have been performed and included in Fig. 8.

We appreciate the Reviewers for providing constructive comments and suggestions for us. We believe the manuscript is now strengthened as a result of the revisions we have made. We hope you agree with us and that you find the paper can be acceptable for publication in Nature Communications.

Sincerely,

Yasuyoshi Ueki MD, PhD

Associate Professor

Dept of Biomedical Sciences and Comprehensive Care, Indiana University School of Dentistry

Indiana Center for Musculoskeletal Health, Indiana University School of Medicine

Van Nuys Medical Science Bldg, Rm#514, 635 Barnhill Dr, Indianapolis, IN 46202

REVIEWERS' COMMENTS

Reviewer #1 (Remarks to the Author):

All comments raised previously have been addressed in a satisfactory manner. The authors should be congratulated for their work which certainly will advance our understanding of how RANKL expression is regulated in periodontal disease.

Reviewer #2 (Remarks to the Author):

Overall this is felt to be an attentive revised manuscript. Although this reviewer feels that there should be more emphasis on use of live bacteria (*Porphyromonas gingivalis*) throughout the array mechanistic assessments where purified ligands were used to drive highly specific process that may not represent what is found when whole bacterial challenge (complex stimulus) is used, there is important specific findings that are felt significant and important when accompanied by targeted studies using live bacteria. No further comments as all my other comments were effectively addressed.

Reviewer #3 (Remarks to the Author):

While the authors have addressed most of my comments, they have essentially ignored the possibility that stromal cells, sometimes referred to as CAR cells, might contribute some or most of the RANKL in their studies. None of the evidence presented by the authors addresses whether RANKL in CAR cells was affected by their conditional deletion or activation using *Dmp1-Cre*. It is possible that the authors are not convinced by published studies that show *Dmp1-Cre* targets CAR cells, or maybe they are not convinced that RANKL deletion from CAR cells causes osteopetrosis. It is also possible that they do not think that CAR cells are present in alveolar bone or the calvarium. If this is the case, then they should at least acknowledge these studies and state that they did not think it was worth exploring the possible contribution of CAR cells.

October 10, 2022

Dear Reviewers,

We sincerely thank all reviewers for reviewing our manuscript again. We carefully considered the comments from the Reviewers. The revisions made in response to the comments of each Reviewer are as follows:

Reviewer #1:

All comments raised previously have been addressed in a satisfactory manner. The authors should be congratulated for their work which certainly will advance our understanding of how RANKL expression is regulated in periodontal disease.

We appreciate the comment. We also thank this Reviewer for carefully reviewing our manuscript. Fruitful suggestions certainly strengthened our conclusion.

Reviewer #2:

Overall this is felt to be an attentive revised manuscript. Although this reviewer feels that there should be more emphasis on use of live bacteria (*Porphyromonas gingivalis*) throughout the array mechanistic assessments where purified ligands were used to drive highly specific process that may not represent what is found when whole bacterial challenge (complex stimulus) is used, there is important specific findings that are felt significant and important when accompanied by targeted studies using live bacteria. No further comments as all my other comments were effectively addressed.

We completely agree with the Reviewer. We also recognize that molecular mechanisms discovered by using Pam3CSK4/LPS need to be confirmed by live bacteria. However, PAMPs of live bacteria is very heterogeneous and complex, and live bacteria certainly express the ligands that stimulate the signaling pathways other than the MYD88 pathway (e. g., TICAM1 (TRIF) pathway, pathways mediated via NOD1/2, C-type lectin receptors, and complement C5a receptor and CXCR4 (for *Pg*)), which dilutes our focus on the MYD88 pathway in osteocytes and makes it difficult to conclude that the RANKL-regulating mechanisms discovered in the current study are solely due to the consequence of the MYD88 pathway activation. Therefore, we decided to use the Pam3CSK4, a “very specific” activator for the MYD88 pathway via TLR2/1, in *in vitro* studies.

Because, in both calvarial injection and oral infection models, *Dmp1-Cre;Myd88^{fl/fl}* mice protect against osteolysis caused by live *Porphyromonas gingivalis*, we think our conclusion that the MYD88 pathway in osteocytes/osteoblasts is critical in bacterial osteolysis is strong and reliable. In future studies, we will examine if these molecular mechanisms can be recapitulated in osteocytes stimulated by live *Porphyromonas gingivalis* or *Escherichia coli*, or *Staphylococcus aureus*.

We hope that the reviewer kindly understands and accepts the reason why we used the purified ligands for *in vitro* studies. The importance of further signaling studies using live bacteria has been discussed in the discussion section as below.

Ultimately, these molecular mechanisms of inducing RANKL expression identified by using Pam3 will need to be validated in osteocytes stimulated with live bacteria.

Reviewer #3:

While the authors have addressed most of my comments, they have essentially ignored the possibility that stromal cells, sometimes referred to as CAR cells, might contribute some or most of the RANKL in their studies. None of the evidence presented by the authors addresses whether RANKL in CAR cells was affected by their conditional deletion or activation using Dmp1-Cre. It is possible that the authors are not convinced by published studies that show Dmp1-Cre targets CAR cells, or maybe they are not convinced that RANKL deletion from CAR cells causes osteopetrosis. It is also possible that they do not think that CAR cells are present in alveolar bone or the calvarium.

If this is the case, then they should at least acknowledge these studies and state that they did not think it was worth exploring the possible contribution of CAR cells.

We sincerely apologize to the reviewer that we did not mention about the CAR cells. We definitely recognize that *Dmp1-Cre* targets a subset of CAR cells as clearly demonstrated in the previous elegant genetic study (Zhang and Link, JBMR 2016, 31, 2001-2007) and the CAR cells differentiate into osteoblasts and adipocytes that express RANKL.

On the other hand, we were not able to find publications showing that the CAR cell itself is a RANKL source for osteoclastogenesis (PubMed searched by CAR cells & RANKL or by CAR cells & osteoclasts). Moreover, Fujiwara et al., a paper from the laboratory of Dr. O'Brien, who is an expert in the RANKL biology, clearly indicated that CAR cells do not supply RANKL for osteoclast formation (J Biol Chem 2016, 291, 48, 24838-24850). Therefore, we decided not to mention about CAR cells in the first revision after serious discussion in our group.

However, we respectfully agree with the Reviewer's concern about the potential impact of CAR cells. Therefore, we have discussed the possible impact of MYD88 deletion/restoration in a subset of CAR cells in this study. Since *Dmp1-Cre* also targets muscle cells and transcortical perivascular cells in proximity to the bone, these cells were also discussed. Further, the importance of investigating if CAR cells can be a RANKL source in bone infection has been mentioned as below.

Importantly, the possible MYD88 deletion/restoration in a subset of CAR cells, skeletal muscle cells, and transcortical perivascular cells by Dmp1-Cre needs to be considered in the interpretation of our findings.^{36-38,82} Particularly, it would be of interest to examine if CAR cells are a critical RANKL supplier for osteoclast induction in bone infection.

Again, we are so sorry that we did not mention about CAR cells. We want the Reviewer to understand that we are not ignoring the critical role of CAR cells and its potential role in bone infection.

We thank the Reviewers for carefully reviewing our revised manuscript again. We hope that you find this manuscript is now acceptable for publication in *Nature Communications*.

Sincerely,

Yasuyoshi Ueki MD, PhD

Associate Professor

Dept of Biomedical Sciences and Comprehensive Care, Indiana University School of Dentistry

Indiana Center for Musculoskeletal Health, Indiana University School of Medicine

Van Nuys Medical Science Bldg, Rm#514, 635 Barnhill Dr, Indianapolis, IN 46202